# Multi-Adapter Representation Interventions via Energy Calibration

**Manjiang Yu** [1 2 3]  **Hongji Li** [2]  **Junwei Chen** [3]  **Xue Li** [1]  **Priyanka Singh** [1]  **Yang Cao** [3]  **Lijie Hu** [2]

## Abstract

Representation intervention has emerged as a promising paradigm for aligning Large Language Models (LLMs) toward desired behaviors without modifying model weights. Existing methods typically apply a fixed intervention uniformly across all inputs. However, we find that the appropriate intervention direction and strength vary substantially across samples, and such indiscriminate intervention leads to the degradation of general capabilities on benign inputs. To address these challenges, we propose **M**ulti-**A**dapter **R**epresentation **I**nterventions via Energy Calibration (**MARI**). Specifically, we introduce a Competitive Multi-Adapter mechanism where specialized experts capture non-linear correction patterns to adaptively determine the appropriate intervention direction and strength for different samples. Furthermore, we design an Energy-Based Gating module that leverages internal propagation dynamics to distinguish inputs applicable for intervention. Extensive experiments across diverse model families and parameter scales demonstrate that MARI achieves state-of-the-art alignment performance. Our method significantly improves TruthfulQA, BBQ, and Safety benchmarks while maintaining and even improving the performance of general tasks such as MMLU and ARC. Our code is available at `https://github.com/V1centNevwake/MARI`.

## 1. Introduction

Large Language Models (LLMs) (Brown et al., 2020; Touvron et al., 2023; OpenAI, 2023) have achieved remarkable success across a broad spectrum of natural language process-

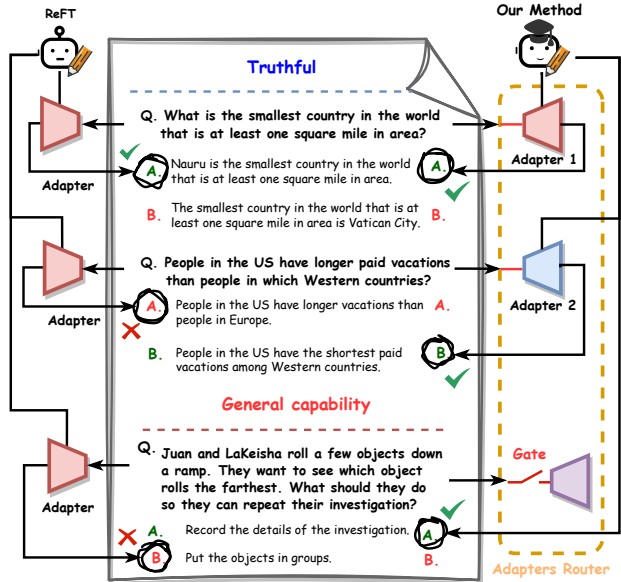

*Figure 1.* **Comparison of intervention results.** (1) **Alignment Reliability:** Static methods (left) rely on a fixed adapter, which leads to unstable performance. In contrast, MARI (right) ensures consistent correctness across diverse inputs. (2) **General Capability Preservation:** Existing methods intervene indiscriminately, which impairs general capability. MARI overcomes this by employing an energy gate that distinguishes intervention-applicable inputs from benign ones, preserving the model's general capabilities.

ing tasks, ranging from question answering to open-ended generation (Su et al., 2023a;b; Cheng et al., 2024; Zhang et al., 2026). However, ensuring that these models behave in alignment with desired attributes such as truthfulness, fairness, or informativeness remains an open challenge, especially in real-world deployments where outputs must satisfy domain-specific or ethical constraints (Zhang et al., 2025; Wang et al., 2025; Guo et al., 2025; Yang et al., 2026; Zhou et al., 2025; Li et al., 2025b; Su et al., 2025). To address this necessity, a growing body of research has converged on representation intervention as a potent control paradigm. By identifying and manipulating specific patterns within the internal activation space at inference time, this approach enables precise guidance of model outputs toward target properties without the need for modifying the underlying model weights (Rimsky et al., 2024; Yang et al., 2024c; Hu et al., 2024). Most existing intervention methods, such as Activation Steering (Turner et al., 2024; Rimsky et al., 2024)

[1]The University of Queensland, Brisbane, Australia [2]Mohamed bin Zayed University of Artificial Intelligence, Abu Dhabi, United Arab Emirates [3]Institute of Science Tokyo, Tokyo, Japan. Correspondence to: Lijie Hu <lijie.hu@mbzuai.ac.ae>.

*Proceedings of the 43rd International Conference on Machine Learning*, Seoul, South Korea. PMLR 306, 2026. Copyright 2026 by the author(s).

and ReFT (Wu et al., 2024), are essentially based on the linear representation hypothesis (Park et al., 2023; Elhage et al., 2021). This hypothesis posits that high-level semantic concepts are encoded as linear directions in the activation space. Based on this, these methods typically apply a fixed steering vector or low-rank updates uniformly to all inputs.

Despite the empirical success of these methods in aligning LLMs, the underlying linear representation hypothesis does not hold universally. (Cunningham et al., 2023) Intuitively, given the vast diversity of input queries and the inherent complexity of attributes like truthfulness, it is hard to imagine that a single, static global intervention can effectively align all inputs. To further dig into the validity of this assumption, we conduct an empirical analysis in Section 4.1. Our results reveal that a unified intervention vector is strictly unable to accommodate all inputs: optimal correction directions are highly heterogeneous and, in many cases, contradict the global trend. Furthermore, even with a precise intervention direction, ensuring the effectiveness of the intervention without compromising the model's general capability remains a key challenge. Blindly intervening in all queries (including those that don't require correction) can lead to over-intervention, disrupting the model's internal representation of benign inputs. Such indiscriminate application has been shown to reduce the robustness of general inference (Wolf et al., 2024).

To address these challenges, we propose **MARI**. To cope with heterogeneous intervention requirements, we introduce a **Competitive Multi-Adapter** mechanism that learns multiple lightweight low-rank adapters at a single injection site. Through competitive training and inference-time routing, MARI produces input-adaptive interventions instead of applying a single global edit to all queries. Furthermore, to preserve general capabilities and avoid over-intervention on benign inputs, we design an **Energy-Based Gate** guided by off-subspace regularization. It leverages propagation-response energy under a small probe update to provide a label-free applicability signal, enabling reliable threshold gating that triggers intervention only when beneficial and otherwise falls back to the frozen base model.

Our main contributions are summarized as follows:

- We identify the limitations of static linear interventions for alignment: required intervention direction and strength are highly input- and state-dependent, and indiscriminate always-on editing can degrade general capabilities.
- We propose a **Competitive Multi-Adapter** strategy that trains complementary low-rank adapters and uses inference-time routing to realize input-adaptive representation interventions.
- We introduce an **Energy-Based Gate** with off-subspace regularization, where propagation-response energy provides a label-free applicability signal for reliable threshold

control at inference time.
- We demonstrate great and consistent improvements on alignment benchmarks across model families and scales, while maintaining or even improving general capability.

## 2. Related Work

**Representation Intervention.** Prior studies have demonstrated that LLMs encode rich semantics within their activation space. Core behaviors, including trustworthiness (Zou et al., 2023; Marks & Tegmark, 2023; Li et al., 2025a), refusal (Arditi et al., 2024), and reasoning (Morishita et al., 2024), have been identified as specific patterns within these internal states. Building on these insights, recent research has established that model outputs can be effectively controlled by directly manipulating these internal representations. For example, Activation Steering (Turner et al., 2024; Subramani et al., 2022; Yu et al., 2026; Jiang et al., 2025; 2026) controls model generation by adding a specific steering vector to the hidden states during inference. Representation Finetuning (Wu et al., 2024) intervenes on hidden representations by learning a low-rank update matrix. Additional related work is provided in Appendix C

## 3. Preliminaries

**Representation Intervention.** Let $\mathbf{f}_\theta$ be a pretrained Transformer language model with parameters $\theta$ kept frozen. For an input token sequence $x = (x_1, \ldots, x_n)$, let $\mathbf{h}_p^{(l)}(x) \in \mathbb{R}^d$ denote the hidden state at layer $l$ and token position $p$. We follow the representation-intervention paradigm (e.g., ReFT (Wu et al., 2024)) and learn an editor while leaving the base model unchanged. We intervene at a single layer–position pair $(l^*, p^*)$ (e.g., the last prompt token). Let $\Phi_\psi : \mathbb{R}^d \to \mathbb{R}^d$ be an intervention function with parameters $\psi$. The intervention *replaces* the hidden state at $(l^*, p^*)$ with

$$\tilde{\mathbf{h}}_{p^*}^{(l^*)}(x) = \Phi_\psi\left(\mathbf{h}_{p^*}^{(l^*)}(x)\right). \qquad (1)$$

The edited state is then propagated through the remaining layers, inducing an intervened distribution $p_{\theta,\psi}(\cdot \mid x)$. We train only $\psi$.

**Low-rank editor.** We parameterize $\Phi_\psi$ as an additive low-rank update with a nonnegative global scale $\gamma \geq 0$:

$$\Phi_\psi(\mathbf{h}) = \mathbf{h} + \gamma\, s_\psi\, \Delta_\psi(\mathbf{h}), \qquad (2)$$

where $s_\psi \geq 0$ is an optional (learnable) scalar scale that can be shared or frozen across adapters in our implementation. We instantiate $\Delta_\psi$ with a rank-$r$ factorization

$$\Delta_\psi(\mathbf{h}) = \mathbf{U}\left(\mathbf{V}^\top \mathbf{h} + \mathbf{b}\right), \qquad \mathbf{U}, \mathbf{V} \in \mathbb{R}^{d \times r},\ \mathbf{b} \in \mathbb{R}^r, \tag{3}$$

so that the intervention output always lies in the $r$-dimensional subspace $\mathrm{span}(\mathbf{U})$, while the input is first projected into an $r$-dimensional bottleneck via $\mathbf{V}^\top \mathbf{h}$. This

yields an input-dependent update while restricting its degrees of freedom to a low-dimensional subspace.

## 4. Methodology

We present **MARI**. We begin by motivating our multi-adapter approach through the limitations of the linear representation hypothesis (Section 4.1). Next, we introduce a competitive training strategy where diverse experts collectively realize input-dependent actuation (Section 4.2). Finally, we describe an energy-based gating mechanism to regulate intervention applicability at inference time (Section 4.3).

### 4.1. Motivation: Limitations of the Linear Representation Hypothesis

Many representation interventions (e.g., steering vectors and ReFT) implicitly assume an edit can be captured by a *single, input-invariant* low-dimensional update at a fixed layer—a fixed direction/subspace applied uniformly across inputs. Under this hypothesis, improving an attribute like truthfulness reduces to learning one global low-rank transformation that shifts hidden states toward a desirable region.

**A diagnostic reveals state-dependent heterogeneity.** We test this hypothesis on TruthfulQA (MC1) using a simple representation-space diagnostic. For questions where the frozen model selects an incorrect option $\hat{y}$ while the correct option is $y^\star$, we extract hidden activations at a fixed layer $l^\star$ and form a per-sample correction vector

$$\Delta(x) = a(x, y^\star) - a(x, \hat{y}) \tag{4}$$

where $a(x, y)$ denotes a pooled activation vector over the answer-option span (details in Appendix A). We then sort samples by a scalar coordinate $t(x)$ derived from the hallucinated state (e.g., $\mathrm{PC1}(a(x, \hat{y}))$) and summarize, in sliding windows along $t(x)$, (i) the required *strength* $s(x) = \|\Delta(x)\|_2$ and (ii) the *directional dispersion* of $\Delta(x)$ (using a cosine-based deviation statistic; see Appendix A). Figure 2 reports window-wise medians with quantile bands.

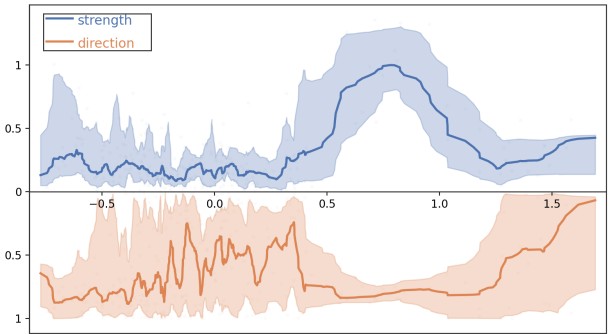

*Figure 2.* **Variability of intervention needs.** The visualization shows that the optimal strength (top) and direction (bottom) vary significantly across different inputs.

**Observation I (State-dependent heterogeneity).** Both the

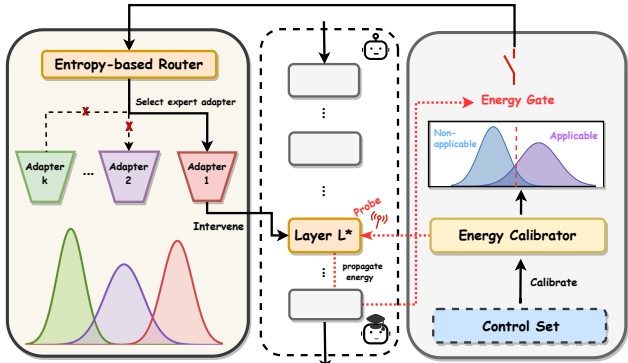

*Figure 3.* **Pipeline of MARI.** MARI integrates an Energy-Based Gate that utilizes propagation dynamics to distinguish intervention-applicable inputs from benign ones. Non-applicable inputs bypass the intervention (falling back to the frozen base model), while applicable inputs are directed by an Entropy Router to one of $K$ competitive experts for precise, input-adaptive steering.

required magnitude and direction of $\Delta(x)$ vary substantially across $t(x)$ (and even locally within windows), indicating that the intervention requirements are not globally consistent; a single global linear edit may therefore fail to cover all regimes reliably.

**Observation II (Over-intervention).** To cover multiple regimes with one edit, one may increase rank or scale, but stronger always-on interventions can perturb benign inputs whose required direction/magnitude differs, degrading general capability. These observations motivate input-adaptive actuation with selective application (routing and gating).

### 4.2. Competitive Multi-Adapters

Motivated by Observation I, we introduce **Competitive Multi-Adapters**: Instead of committing to one global edit, we train multiple lightweight low-rank adapters at the same injection site and encourage them to specialize in competitive training and routing, so that the intervention can adapt to heterogeneous requirements.

We place a set of $K$ ($K \geq 1$) low-rank adapters $\{A_k\}_{k=1}^K$ at a single injection site $(l^\star, p^\star)$ in a frozen LLM $f_\theta$. Adapter $A_k$ parameterized an intervention function $\Phi_{\psi_k}$ (cf. Section 3), which replaces the hidden state $h := h_{p^\star}^{(l^\star)}(x) \in \mathbb{R}^d$ with

$$\tilde{h} = \Phi_{\psi_k}(h) = h + \gamma\, s_k\, \Delta_{\psi_k}(h), \tag{5}$$

where $\gamma \geq 0$ is a global scale and $s_k \geq 0$ is an optional per-adapter scalar. Unless otherwise stated, we fix $s_k \equiv 1$ and use a global actuation scale $\gamma$ shared across all adapters.

We instantiate the low-rank update with a rank-$r$ factorization ( Eq.(3)):

$$\Delta_{\psi_k}(h) = U_k\big(V_k^\top h + b_k\big), \qquad U_k, V_k \in \mathbb{R}^{d \times r},\; b_k \in \mathbb{R}^r, \tag{6}$$

so the intervention output lies in the $r$-dimensional output subspace $\mathrm{span}(U_k)$, while $V_k$ defines an input projection into an $r$-dimensional bottleneck. Each adapter induces an intervened distribution $p_{\theta,\psi_k}(\cdot \mid x)$; we train only $\{\psi_k\}_{k=1}^{K}$ (and optionally $\{s_k\}$), keeping $\theta$ frozen.)

Given a supervised example $(x, y)$, we define a per-adapter loss $\ell_k(x, y)$ that generalizes across both **multiple-choice** and **free-form generation** paradigms. Specifically, we employ the cross-entropy loss over candidate options for multiple-choice tasks, and the teacher-forced negative log-likelihood for generation tasks (see Appendix A, Eqs. 20–21 for formal definitions). To encourage specialization rather than training all adapters toward the same averaged optimum, we perform hard routing during training (winner-take-gradient): each example is assigned to the adapter that currently achieves the lowest supervised loss,

$$k^{\star}(x, y) := \arg\min_{k \in \{1,\ldots,K\}} \ell_k(x, y), \qquad (7)$$

and the task loss is backpropagated only through the winner adapter:

$$\mathcal{L}_{\text{route}} := \mathbb{E}_{(x,y)}\big[ \ell_{k^{\star}(x,y)}(x, y) \big]. \qquad (8)$$

In practice, we implement Eq.(8) by detaching non-winner losses, so gradients from the supervised objective flow only to $\psi_{k^{\star}}$. To prevent mode collapse under hard routing, we use a lightweight minibatch-level usage balancing term.

**Entropy routing induces piecewise-affine actuation** At inference time, the oracle routing in Eq.(7) is unavailable since $y$ is unknown. We therefore select an adapter using a parameter-free uncertainty criterion based on predictive entropy. Because all experts share the same frozen backbone and output head, we compute uncertainty scores using a shared softmax temperature (no per-expert temperature or logit scaling), so the entropy $u_k(x)$ are directly comparable across $k$. We use the entropy functional

$$H(p) := -\sum_j p_j \log p_j. \qquad (9)$$

We compute an uncertainty score $u_k(x)$ from the predictive distribution under adapter $k$ (e.g., option entropy for multiple-choice, or averaged next-token entropy for generation; see App. Section A, Eqs. (27)–(28)), and route by selecting the most confident adapter:

$$\hat{k}(x) := \arg\min_{k \in \{1,\ldots,K\}} u_k(x). \qquad (10)$$

This routing is training-free, introduces no additional parameters, and keeps $K$ as a simple hyperparameter.

Let $\pi(x) \in \{1, \ldots, K\}$ denote the routing rule, using training-time oracle routing in Eq.(7) and inference-time

entropy routing in Eq.(10). It induces a partition of the input space $\mathcal{R}_k := \{x : \pi(x) = k\}$. The overall actuation update can be written as

$$\delta(x) := \Delta_{\psi_{\pi(x)}}\big(h(x)\big), \qquad (11)$$

so that on each region $\mathcal{R}_k$ the intervention reduces to the rank-$r$ affine map in Eq. (6), while input-dependence arises solely from $\pi(x)$.

### 4.3. Energy-based Gate

Applying interventions indiscriminately risks over-steering and can harm benign inputs, motivating an input-wise criterion for **when** to intervene. Recent work shows that internal model signals can selectively trigger or scale interventions to improve the alignment–utility trade-off. (Hedström et al., 2025; Ferrando et al., 2025) Motivated by these findings, we introduce an energy-based gate that uses propagation-response dynamics under a small probe update to compute a label-free applicability score, and applies thresholding to trigger actuation only when beneficial, while otherwise falling back to the frozen base model.

Specifically, fix an injection site $(l^{\star}, p^{\star})$. Let $\{h_{p^{\star}}^{(m)}(x)\}_{m=l^{\star}}^{L}$ denote the hidden states of the frozen model on input $x$ (no injection). Given a probe injection of strength $\alpha$, let $\{h_{p^{\star}}^{(\alpha,m)}(x)\}_{m=l^{\star}}^{L}$ be the corresponding hidden states under the probe run. We define the per-layer propagation response $e_m(x; \alpha) \triangleq \|h_{p^{\star}}^{(\alpha,m)}(x) - h_{p^{\star}}^{(m)}(x)\|_2$ (see Eq. (29) in App. Section A) and summarize it by a robust statistic

$$E(x; \alpha) \triangleq \mathrm{median}\Big(\{e_m(x; \alpha)\}_{m=l^{\star}}^{L}\Big). \qquad (12)$$

Intuitively, $E(x; \alpha)$ measures how strongly a small injected update propagates through the remaining network; we use it as a proxy for whether applying the full intervention is likely to behave stably on input $x$.

To obtain a comparable and intervention-relevant energy signal, we realize the probe with a low-rank module $g_\phi$ (Energy Calibrator), drawn from the same intervention family as Section 4.2 but with a smaller rank $r_{\text{probe}}$. Unlike the actuation experts, $g_\phi$ is trained independently and is used only to generate probe updates (never for generation). Given $h(x) = h_{p^{\star}}^{(l^{\star})}(x)$, the calibrator produces a probe update

$$\delta_\phi(x) \triangleq g_\phi(h(x)), \qquad (13)$$

and we compute $E(x; \alpha_{\text{probe}})$ by injecting $\alpha_{\text{probe}}\, \delta_\phi(x)$ during the probe run. Using a single dedicated probe avoids coupling control to expert routing and prevents the actuation experts from being constrained to also produce a convenient gating signal.

**Threshold gating with base-model fallback.** We train the probe $g_\phi$ on in-field supervision using the same per-example task losses as in Section 4.2 (Eqs. (20)–(21)), but without competitive routing. Let $\ell_\phi(x, y)$ denote the task loss evaluated under the probe-injected model $p_{\theta,\phi}(\cdot \mid x)$. We optimize

$$\mathcal{L}_{\text{cal}} := \mathbb{E}_{(x,y) \sim \mathcal{D}_{\text{in}}}[\ell_\phi(x, y)] + \lambda_{\text{off}} \mathcal{R}_{\text{off}}, \qquad (14)$$

where $\mathcal{R}_{\text{off}} := \mathbb{E}_{x \sim \mathcal{D}_{\text{in}}^{\text{pca}}}[\mathcal{L}_{\text{off}}(x)]$ is an off-subspace regularizer computed on an *unlabeled* subset $\mathcal{D}_{\text{in}}^{\text{pca}} \subset \mathcal{D}_{\text{in}}$ (inputs only; labels are ignored).

**Off-subspace penalty improves separability.** We construct an in-field calibration subspace at the injection layer by fitting a rank-$k$ PCA basis $B$ on $\mathcal{D}_{\text{in}}^{\text{pca}}$ and penalize the probe component outside this subspace:

$$\mathcal{L}_{\text{off}}(x) \triangleq \left\| \Pi_B^\perp(\delta_\phi(x)) \right\|_2^2. \qquad (15)$$

(Definitions of $\Pi_B^\perp$ and the PCA construction are standard; we include the exact operators in App. Section A, Eq. (30).) This shapes the probe to produce in-field-compatible updates at the injection layer and empirically improves the separability of $E(\cdot)$ between inputs where actuation is applicable vs. non-applicable.

We convert energy into a binary applicability decision by calibrating a threshold $\tau_E$ on a small held-out control set $\mathcal{D}_{\text{ctrl}}$ that contains both applicable and non-applicable inputs. We compute the energy scores $\{E(x; \alpha_{\text{probe}})\}_{x \in \mathcal{D}_{\text{ctrl}}}$ and choose $\tau_E$ by a fixed operating rule: let $\mathcal{D}_{\text{ctrl}}^{\text{non}} \subset \mathcal{D}_{\text{ctrl}}$ denote the non-applicable portion, and pick $\tau_E$ as the $(1 - \rho)$-quantile of $\{E(x; \alpha_{\text{probe}})\}_{x \in \mathcal{D}_{\text{ctrl}}^{\text{non}}}$, so that $\Pr_{x \sim \mathcal{D}_{\text{ctrl}}^{\text{non}}}[E(x; \alpha_{\text{probe}}) \geq \tau_E] \approx \rho$ (i.e., we reject/intercept a target fraction $\rho$ of non-applicable inputs). In our experiments, $\rho$ is a hyperparameter, and we typically set $\rho = 0.9$.

At inference time, we set the actuation strength via

$$\alpha(x) := \begin{cases} \alpha_{\text{full}}, & E(x; \alpha_{\text{probe}}) \geq \tau_E \quad \text{(applicable)}, \\ \alpha_{\text{safe}}, & E(x; \alpha_{\text{probe}}) < \tau_E \quad \text{(non-applicable)}, \end{cases} \qquad (16)$$

where we typically set $\alpha_{\text{safe}} = 0$ (suppress actuation), falling back to the frozen base model. When $\alpha(x) = \alpha_{\text{full}}$,

## 5. Theoretical Analysis

Detailed definitions, auxiliary lemmas, assumptions, propositions, and proofs are deferred to Appendix B.

**Theorem 5.1** (**Risk Bound and Improvement Condition**). *Let loss* $\ell_k : \mathcal{X} \times \mathcal{Y} \to [0, L]$ *be bounded. For each example* $(x, y)$, *let the oracle expert be* $k^\star(x, y) := \arg\min_k \ell_k(x, y)$ *and the entropy-selected expert be*

$\hat{k}(x) := \arg\min_k u_k(x)$, *where* $u_k(x)$ *is the entropy-based uncertainty score used at inference. Define the oracle risk* $R_{\min} \triangleq \mathbb{E}[\ell_{k^\star}]$ *and the misrouting rate* $\eta \triangleq \Pr(\hat{k} \neq k^\star)$. *The risk of entropy routing,* $R_{\text{ent}}$, *satisfies:*

$$R_{\text{ent}} \leq R_{\min} + L \cdot \eta. \qquad (17)$$

*Defining the specialization gain* $\Delta_{\text{spec}} \triangleq R_{\text{single}} - R_{\min}$, *we derive the sufficiency condition for improvement:*

$$\Delta_{\text{spec}} > L \cdot \eta \implies R_{\text{ent}} < R_{\text{single}}. \qquad (18)$$

> **Remark**
>
> Eq. (17) makes the routing–specialization trade-off explicit: gains come from $\Delta_{\text{spec}}$, while imperfect routing incurs a penalty proportional to $\eta$. EG-MARI increases $\Delta_{\text{spec}}$ via competitive specialization and keeps $\eta$ small via a calibrated, label-free uncertainty criterion at inference. Since the competitive loss is on a shared scale across experts, the single-adapter entropy-to-correctness accuracy admits a consistent lower bound; see Appendix B.2.

**Theorem 5.2** (**Upper Bound on Non-Applicable Energy**). *Consider the propagation energy* $E(x; \alpha)$ *and the in-field subspace* $B$. *Suppose the probe* $\delta_\phi$ *satisfies the alignment constraint* $\|\Pi_B^\perp \delta_\phi\|_2 \leq \varepsilon$ *and bounded magnitude* $\|\Pi_B \delta_\phi\|_2 \leq S$. *Assume that for non-applicable inputs* $x \sim \mathcal{D}_{\text{non}}$, *the downstream propagation gain along* $B$ *is attenuated by* $\kappa_{\text{non}}$ *(Assumption C.8). Under a first-order sensitivity model, the energy is bounded by:*

$$E(x; \alpha) \leq \alpha \cdot \underbrace{(\kappa_{\text{non}} S + \Gamma(x) \varepsilon)}_{\triangleq C_{\text{non}}(x)} + o(\alpha), \qquad (19)$$

*where* $\Gamma(x)$ *denotes the spectral norm of the Jacobian restricted to the orthogonal complement* $B^\perp$. *Thus,* $E(x; \alpha)$ *vanishes linearly as* $\alpha \to 0$, *with a coefficient determined by in-field attenuation* $\kappa_{\text{non}}$ *and the probe's off-subspace leakage* $\varepsilon$.

> **Remark**
>
> Eq. (19) decomposes low non-applicable energy into two factors: (i) *subspace alignment* (small $\varepsilon$) reduces the $B^\perp$ leakage term weighted by $\Gamma(x)$, and (ii) *attenuated in-field propagation* (small $\kappa_{\text{non}}$) weakens downstream response along $B$. Together, small $\kappa_{\text{non}}$ and $\varepsilon$ tighten the bound, making non-applicable inputs tend to yield low energy.

*Table 1.* Evaluation results on alignment benchmarks (TruthfulQA, BBQ, Refusal) and general capabilities (MMLU, ARC). Results are reported as mean$_{\pm\text{std}}$. **MARI** achieves the best trade-off, consistently ranking first or second across all metrics. The best results are **bolded** and the second-best underlined.

| Method | TruthfulQA | | Bias | Safety | General Capabilities | | |
|---|---|---|---|---|---|---|---|
| | MC1 ($\uparrow$) | MC2 ($\uparrow$) | BBQ ($\uparrow$) | Refusal ($\uparrow$) | MMLU ($\uparrow$) | ARC-E ($\uparrow$) | ARC-C ($\uparrow$) |
| *Llama-2-7B* | | | | | | | |
| Vanilla | $32.03_{\pm0.21}$ | $49.51_{\pm0.28}$ | $0.329_{\pm0.011}$ | $0.821_{\pm0.021}$ | $23.3_{\pm0.3}$ | $51.9_{\pm0.4}$ | $33.8_{\pm0.3}$ |
| CAA | $33.80_{\pm0.18}$ | $46.90_{\pm0.23}$ | $0.515_{\pm0.009}$ | $0.830_{\pm0.018}$ | $23.1_{\pm0.4}$ | $\underline{52.2}_{\pm0.3}$ | $33.5_{\pm0.4}$ |
| ITI | $30.81_{\pm0.20}$ | $49.80_{\pm0.29}$ | $0.520_{\pm0.012}$ | $0.823_{\pm0.015}$ | $\mathbf{23.6}_{\pm0.3}$ | $51.7_{\pm0.5}$ | $\mathbf{34.1}_{\pm0.3}$ |
| NL-ITI | $31.30_{\pm0.25}$ | $49.38_{\pm0.26}$ | $0.522_{\pm0.013}$ | $0.840_{\pm0.024}$ | $\underline{23.4}_{\pm0.2}$ | $52.0_{\pm0.4}$ | $33.7_{\pm0.3}$ |
| ReFT | $\underline{50.46}_{\pm0.33}$ | $\underline{50.20}_{\pm0.31}$ | $\underline{0.540}_{\pm0.015}$ | $\underline{0.845}_{\pm0.020}$ | $23.2_{\pm0.4}$ | $51.8_{\pm0.3}$ | $\underline{34.0}_{\pm0.4}$ |
| **MARI** | $\mathbf{64.35}_{\pm0.12}$ | $\mathbf{61.13}_{\pm0.15}$ | $\mathbf{0.751}_{\pm0.008}$ | $\mathbf{0.851}_{\pm0.011}$ | $23.2_{\pm0.2}$ | $\mathbf{52.6}_{\pm0.3}$ | $33.5_{\pm0.2}$ |
| *Llama-3-8B* | | | | | | | |
| Vanilla | $28.70_{\pm0.21}$ | $45.03_{\pm0.28}$ | $0.608_{\pm0.011}$ | $0.851_{\pm0.021}$ | $65.9_{\pm0.3}$ | $77.3_{\pm0.4}$ | $51.4_{\pm0.3}$ |
| CAA | $29.64_{\pm0.18}$ | $46.95_{\pm0.23}$ | $0.635_{\pm0.009}$ | $0.854_{\pm0.018}$ | $65.5_{\pm0.2}$ | $\underline{77.7}_{\pm0.5}$ | $51.1_{\pm0.4}$ |
| ITI | $35.45_{\pm0.20}$ | $\underline{53.95}_{\pm0.29}$ | $0.612_{\pm0.012}$ | $0.813_{\pm0.015}$ | $\underline{66.1}_{\pm0.4}$ | $77.2_{\pm0.3}$ | $\underline{51.7}_{\pm0.3}$ |
| NL-ITI | $36.19_{\pm0.25}$ | $53.12_{\pm0.26}$ | $0.625_{\pm0.013}$ | $0.820_{\pm0.024}$ | $65.8_{\pm0.3}$ | $77.4_{\pm0.4}$ | $51.3_{\pm0.3}$ |
| ReFT | $\underline{50.58}_{\pm0.33}$ | $52.51_{\pm0.31}$ | $\underline{0.637}_{\pm0.015}$ | $\underline{0.861}_{\pm0.020}$ | $66.0_{\pm0.3}$ | $77.1_{\pm0.5}$ | $51.6_{\pm0.4}$ |
| **MARI** | $\mathbf{61.81}_{\pm0.14}$ | $\mathbf{67.39}_{\pm0.18}$ | $\mathbf{0.792}_{\pm0.007}$ | $\mathbf{0.866}_{\pm0.010}$ | $\mathbf{66.6}_{\pm0.2}$ | $\mathbf{78.0}_{\pm0.2}$ | $\mathbf{52.1}_{\pm0.3}$ |
| *Llama-2-13B* | | | | | | | |
| Vanilla | $32.76_{\pm0.22}$ | $50.75_{\pm0.27}$ | $0.333_{\pm0.012}$ | $0.714_{\pm0.022}$ | $26.0_{\pm0.3}$ | $53.7_{\pm0.4}$ | $42.6_{\pm0.3}$ |
| CAA | $33.50_{\pm0.19}$ | $51.90_{\pm0.24}$ | $0.628_{\pm0.010}$ | $0.720_{\pm0.019}$ | $25.7_{\pm0.4}$ | $\mathbf{53.9}_{\pm0.5}$ | $42.3_{\pm0.4}$ |
| ITI | $35.45_{\pm0.23}$ | $\underline{53.95}_{\pm0.30}$ | $\underline{0.712}_{\pm0.014}$ | $0.265_{\pm0.028}$ | $\underline{26.3}_{\pm0.3}$ | $53.5_{\pm0.4}$ | $\mathbf{42.9}_{\pm0.3}$ |
| NL-ITI | $36.19_{\pm0.26}$ | $53.12_{\pm0.28}$ | $0.630_{\pm0.011}$ | $0.715_{\pm0.025}$ | $26.1_{\pm0.2}$ | $\underline{53.8}_{\pm0.3}$ | $42.5_{\pm0.4}$ |
| ReFT | $\underline{59.80}_{\pm0.35}$ | $55.60_{\pm0.32}$ | $0.678_{\pm0.016}$ | $\underline{0.728}_{\pm0.021}$ | $25.9_{\pm0.3}$ | $53.6_{\pm0.4}$ | $\underline{42.8}_{\pm0.3}$ |
| **MARI** | $\mathbf{66.67}_{\pm0.16}$ | $\mathbf{64.96}_{\pm0.19}$ | $\mathbf{0.784}_{\pm0.009}$ | $\mathbf{0.738}_{\pm0.012}$ | $\mathbf{26.7}_{\pm0.2}$ | $53.4_{\pm0.3}$ | $42.4_{\pm0.2}$ |
| *Qwen2-7B* | | | | | | | |
| Vanilla | $26.75_{\pm0.20}$ | $48.31_{\pm0.29}$ | $0.638_{\pm0.011}$ | $0.631_{\pm0.019}$ | $72.4_{\pm0.3}$ | $75.5_{\pm0.4}$ | $51.9_{\pm0.3}$ |
| CAA | $28.44_{\pm0.17}$ | $49.25_{\pm0.25}$ | $0.665_{\pm0.008}$ | $0.644_{\pm0.017}$ | $72.1_{\pm0.4}$ | $\mathbf{75.8}_{\pm0.5}$ | $51.6_{\pm0.4}$ |
| ITI | $31.12_{\pm0.22}$ | $52.32_{\pm0.31}$ | $0.668_{\pm0.013}$ | $0.635_{\pm0.021}$ | $\underline{72.6}_{\pm0.3}$ | $75.3_{\pm0.4}$ | $\underline{52.2}_{\pm0.3}$ |
| NL-ITI | $32.84_{\pm0.24}$ | $52.33_{\pm0.27}$ | $\underline{0.683}_{\pm0.010}$ | $0.633_{\pm0.023}$ | $72.5_{\pm0.2}$ | $\underline{75.6}_{\pm0.3}$ | $51.8_{\pm0.4}$ |
| ReFT | $\underline{49.83}_{\pm0.34}$ | $\underline{59.69}_{\pm0.30}$ | $0.678_{\pm0.015}$ | $\underline{0.646}_{\pm0.018}$ | $72.3_{\pm0.4}$ | $75.4_{\pm0.5}$ | $52.1_{\pm0.3}$ |
| **MARI** | $\mathbf{65.74}_{\pm0.11}$ | $\mathbf{68.84}_{\pm0.14}$ | $\mathbf{0.784}_{\pm0.006}$ | $\mathbf{0.655}_{\pm0.010}$ | $\mathbf{73.1}_{\pm0.2}$ | $75.2_{\pm0.2}$ | $\mathbf{52.3}_{\pm0.2}$ |
| *Qwen2.5-14B* | | | | | | | |
| Vanilla | $39.81_{\pm0.25}$ | $60.71_{\pm0.28}$ | $0.648_{\pm0.011}$ | $0.626_{\pm0.020}$ | $80.9_{\pm0.3}$ | $84.7_{\pm0.4}$ | $63.4_{\pm0.3}$ |
| CAA | $40.44_{\pm0.21}$ | $62.25_{\pm0.24}$ | $0.655_{\pm0.009}$ | $0.634_{\pm0.018}$ | $80.6_{\pm0.4}$ | $\underline{85.0}_{\pm0.5}$ | $63.1_{\pm0.4}$ |
| ITI | $42.20_{\pm0.26}$ | $63.65_{\pm0.32}$ | $0.657_{\pm0.012}$ | $0.632_{\pm0.022}$ | $\underline{81.2}_{\pm0.3}$ | $84.5_{\pm0.4}$ | $\underline{63.8}_{\pm0.3}$ |
| NL-ITI | $46.84_{\pm0.28}$ | $\underline{68.33}_{\pm0.29}$ | $\underline{0.673}_{\pm0.011}$ | $\underline{0.640}_{\pm0.023}$ | $81.0_{\pm0.2}$ | $84.8_{\pm0.3}$ | $63.3_{\pm0.4}$ |
| ReFT | $\underline{52.33}_{\pm0.36}$ | $62.80_{\pm0.34}$ | $0.646_{\pm0.015}$ | $0.636_{\pm0.020}$ | $80.8_{\pm0.4}$ | $84.6_{\pm0.5}$ | $63.6_{\pm0.3}$ |
| **MARI** | $\mathbf{67.93}_{\pm0.13}$ | $\mathbf{71.92}_{\pm0.16}$ | $\mathbf{0.821}_{\pm0.007}$ | $\mathbf{0.645}_{\pm0.011}$ | $\mathbf{81.6}_{\pm0.2}$ | $\mathbf{85.4}_{\pm0.2}$ | $\mathbf{64.1}_{\pm0.2}$ |
| *Qwen2.5-32B* | | | | | | | |
| Vanilla | $43.98_{\pm0.24}$ | $59.81_{\pm0.29}$ | $0.736_{\pm0.012}$ | $0.621_{\pm0.021}$ | $83.5_{\pm0.3}$ | $77.8_{\pm0.4}$ | $59.3_{\pm0.3}$ |
| CAA | $44.77_{\pm0.20}$ | $60.05_{\pm0.26}$ | $0.746_{\pm0.009}$ | $\mathbf{0.663}_{\pm0.018}$ | $83.2_{\pm0.4}$ | $\underline{78.1}_{\pm0.5}$ | $59.0_{\pm0.4}$ |
| ITI | $46.78_{\pm0.28}$ | $64.43_{\pm0.33}$ | $0.755_{\pm0.013}$ | $0.645_{\pm0.022}$ | $\underline{83.7}_{\pm0.3}$ | $77.6_{\pm0.4}$ | $\underline{59.6}_{\pm0.3}$ |
| NL-ITI | $47.92_{\pm0.29}$ | $\underline{66.23}_{\pm0.30}$ | $0.772_{\pm0.011}$ | $0.642_{\pm0.024}$ | $83.6_{\pm0.2}$ | $77.9_{\pm0.3}$ | $59.2_{\pm0.4}$ |
| ReFT | $\underline{55.60}_{\pm0.39}$ | $48.90_{\pm0.35}$ | $\underline{0.821}_{\pm0.016}$ | $0.648_{\pm0.019}$ | $83.4_{\pm0.4}$ | $77.7_{\pm0.5}$ | $59.5_{\pm0.3}$ |
| **MARI** | $\mathbf{81.94}_{\pm0.15}$ | $\mathbf{70.59}_{\pm0.18}$ | $\mathbf{0.876}_{\pm0.008}$ | $\underline{0.653}_{\pm0.010}$ | $\mathbf{84.2}_{\pm0.2}$ | $\mathbf{78.5}_{\pm0.2}$ | $\mathbf{60.0}_{\pm0.2}$ |

## 6. Experiments

### 6.1. Experimental Settings

Detailed information regarding the dataset, evaluation protocols, baseline, and implementation settings is provided in Appendix A.

**Models.** We conduct experiments across a diverse set of state-of-the-art open-weights LLMs, covering different model families and parameter scales ranging from 7B to 32B. Specifically, we examine the Llama family, including Llama-2 (7B, 13B) (Touvron et al., 2023) and Llama-3-8B (Grattafiori et al., 2024), as well as the Qwen family, including Qwen2-7B (Yang et al., 2024a) and Qwen2.5 models (14B, 32B) (Yang et al., 2024b).

**Datasets.** We evaluate alignment efficacy on three widely

*Table 2.* **Ablation Study.** We analyze the contribution of key components in EG-MARI. Note: While removing Energy Gating often yields higher alignment scores, it causes severe regression in general capabilities. EG-MARI offers the best balance.

| Method | TruthfulQA | | Bias | Safety | General Capabilities | | |
| --- | --- | --- | --- | --- | --- | --- | --- |
| | MC1 ($\uparrow$) | MC2 ($\uparrow$) | BBQ ($\uparrow$) | Refusal ($\uparrow$) | MMLU ($\uparrow$) | ARC-E ($\uparrow$) | ARC-C ($\uparrow$) |
| *Llama-3-8B* | | | | | | | |
| Vanilla | 28.70 | 45.03 | 0.608 | 0.851 | 65.9 | 77.3 | 51.4 |
| w/o Energy Gating | **65.15** | **68.20** | **0.800** | 0.860 | 57.5 | 70.2 | 44.8 |
| w/o Multi-Adapter | 45.80 | 55.40 | 0.680 | 0.831 | 66.2 | 77.8 | 51.8 |
| **EG-MARI** | 61.81 | 67.39 | 0.792 | **0.866** | **66.6** | **78.0** | **52.1** |
| *Qwen2.5-14B* | | | | | | | |
| Vanilla | 39.81 | 60.71 | 0.648 | 0.626 | 80.9 | 84.7 | 63.4 |
| w/o Energy Gating | 71.50 | 70.10 | **0.828** | **0.665** | 74.5 | 78.1 | 58.2 |
| w/o Multi-Adapter | 52.30 | 67.80 | 0.640 | 0.620 | 81.2 | 85.1 | 63.9 |
| **EG-MARI** | **67.93** | **71.92** | 0.821 | 0.645 | **81.6** | **85.4** | **64.1** |
| *Qwen2.5-32B* | | | | | | | |
| Vanilla | 43.98 | 59.81 | 0.736 | 0.621 | 83.5 | 77.8 | 59.3 |
| w/o Energy Gating | **82.10** | **73.80** | 0.875 | **0.675** | 76.2 | 72.4 | 52.5 |
| w/o Multi-Adapter | 55.60 | 58.90 | 0.640 | 0.635 | 83.9 | 78.2 | 59.8 |
| **EG-MARI** | 81.94 | 70.59 | **0.876** | 0.653 | **84.2** | **78.5** | **60.0** |

used benchmarks: **TruthfulQA** (Lin et al., 2022) for truthfulness, **BBQ** (Parrish et al., 2022) for social bias, and **Sorry-Bench** (Xie et al., 2025) for safety refusal. Additionally, to verify the preservation of general capabilities, we utilize **MMLU** (Hendrycks et al., 2020), **ARC-Easy**, and **ARC-Challenge** (Clark et al., 2018).

**Metrics.** For TruthfulQA, we report MC1 and MC2 accuracies for multiple-choice tasks. For BBQ and general benchmarks (MMLU, ARC), we report standard Accuracy. For safety, we measure the Refusal Rate (Sorry-rate) using a model-based judge.

**Baselines.** We compare MARI against state-of-the-art representation intervention paradigms, including **CAA** (Rimsky et al., 2024), **ITI** (Li et al., 2023), **NL-ITI** (Hoscilowicz et al., 2024), and **ReFT** (Wu et al., 2024).

## 6.2. Main Results

We present the comprehensive evaluation results in Table 1. We have the following observations:

**State-of-the-art alignment performance.** Our method consistently outperforms existing representation intervention baselines across all evaluated metrics. On TruthfulQA, MARI achieves substantial gains, significantly surpassing prior approaches. Simultaneously, for Safety (Refusal) and Bias (BBQ) benchmarks, our method effectively preserves or even improves performance, avoiding the degradation commonly observed in baseline methods. We attribute these gains to our Competitive Multi-Adapter mechanism, which resolves the directional conflicts inherent in static interventions by utilizing specialized experts for precise steering.

**Preservation of general capabilities.** Unlike prior methods that may degrade general performance due to indis-

criminate intervention, MARI preserves and often improves general capabilities by activating edits only when inputs are intervention-applicable. This behavior is governed by our *Energy-Based Gating* module, which leverages propagation-response dynamics to suppress unnecessary interventions on non-applicable inputs. The gating threshold $\tau_E$ is calibrated once on a small, disjoint control set combining an intervention-attribute dataset with a subset of a general-capability dataset, with both sources providing applicable and non-applicable examples, and then kept fixed across all benchmarks to enable consistent gains without over-intervention. Full construction, labeling, and default PCA/probe ranks are given in Appendix .1.

**Consistent effectiveness across model scales and generations.** The improvements achieved by MARI are robust across diverse model generations and parameter sizes. Notably, our method demonstrates effective scalability with model capacity, yielding significantly more pronounced gains on larger models compared to smaller ones. We hypothesize that this scalability stems from the richer latent knowledge encoded in high-parameter models, which possess more structured representation spaces that allow our method to leverage feature separability more effectively.

## 6.3. Ablation Study

We conduct an ablation study across models of different parameter sizes, results summarized in Table 2.

**(1) w/o Energy Gating.** Removing the energy-based control implies that the intervention is applied indiscriminately to all inputs without distinction. We observe that this yields slightly higher scores on specific alignment metrics. However, this marginal gain comes at the cost of a degradation in general capabilities.

**(2) w/o Multi-Adapter.** Replacing the competitive multi-adapter strategy with a single adapter leads to a drastic decline in alignment efficacy. In contrast, the multi-adapter mechanism achieves superior performance by leveraging competitive experts to adaptively capture the heterogeneous intervention requirements across diverse samples.

## 6.4. Hyperparameter Analysis

We analyze the sensitivity of MARI to two key hyperparameters: the number of competitive adapters $K$ and the target rejection rate $\rho$ for the energy gate.

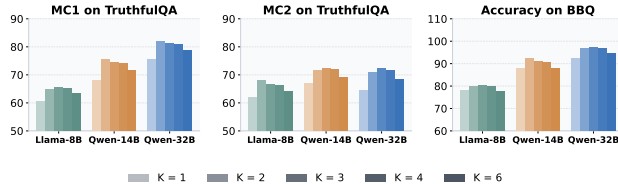

*Figure 4.* **Sensitivity to the number of adapters $K$.** Performance saturates at a small number of experts, with no further improvements observed for larger $K$.

**Impact of adapter count $K$.** As illustrated in Figure 4, the shift from a single adapter to multi-adapter delivers consistent performance gains across all benchmarks. The performance ceiling is typically reached with a small number of experts. Increasing $K$ beyond this point does not yield further improvements and may even lead to regression. We attribute this to the dynamics of competitive training: an excessive number of experts dilutes the training signal received by each adapter, preventing them from learning robust, specialized representations.

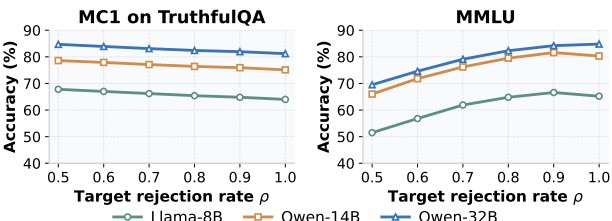

*Figure 5.* **Sensitivity to target rejection rate $\rho$.** Performance remains stable across a wide range of thresholds, indicating that MARI is insensitive to specific hyperparameter settings while consistently balancing alignment and general capabilities.

**Impact of target rejection rate $\rho$.** The target rejection rate $\rho$ governs the aggressiveness of the energy gate. As illustrated in Figure 5, we observe that MARI is remarkably robust to variations in this threshold. Across a broad range, the alignment efficacy (TruthfulQA) remains consistently high without significant fluctuations, while general capabilities (MMLU) are effectively preserved.

*Table 3.* **Fixed-threshold transfer under domain shift.** The threshold $\tau_E$ is calibrated once on $\mathcal{D}_{\text{ctrl}}$ and reused without benchmark-specific recalibration. $\Delta$ denotes the absolute change relative to the vanilla model.

| Setting | Target alignment | | Shifted capability | |
| --- | --- | --- | --- | --- |
| | TQA MC1 | $\Delta$ | GSM8K | $\Delta$ |
| Vanilla | 28.70 | – | 77.4 | – |
| $\rho = 0.50$ | 65.81 | +37.11 | 74.5 | -2.9 |
| $\rho = 0.70$ | 65.35 | +36.65 | 76.8 | -0.6 |
| $\rho = 0.90$ | **64.81** | **+36.11** | **77.4** | **+0.0** |
| $\rho = 1.00$ | 63.90 | +35.20 | 77.2 | -0.2 |

**Fixed-threshold transfer under domain shift.** We further evaluate whether the calibrated energy threshold remains reliable under distribution shift. In all experiments, the threshold $\tau_E$ is calibrated once on the held-out control set $\mathcal{D}_{\text{ctrl}}$ and then kept fixed, without benchmark-specific or test-time recalibration. To stress-test this fixed-threshold setting, we transfer the same threshold to GSM8K, whose distribution differs substantially from the control set and the intervention target domain. As shown in Table 3, MARI maintains strong gains on TruthfulQA while preserving GSM8K performance close to the vanilla model across a wide range of target rejection rates. This suggests that the energy gate is not overly brittle to moderate cross-domain shift, although we do not claim invariance to arbitrary out-of-distribution inputs.

## 6.5. More Analysis

**Complementarity of Multi-Adapter.** To verify that our multi-adapter learn diverse steering patterns rather than converging to a single trivial solution, we analyze the projection of intervention updates $\Delta h$ onto the global mean direction $v_{\text{global}}$ (Figure 6a). We observe that different adapters exhibit distinct distribution profiles with specific mean shifts and spreads. This variation indicates that they capture different intervention patterns to cooperatively cover different regimes of the intervention space, which allows MARI to adaptively address heterogeneous alignment requirements across diverse inputs.

**Energy dynamics provide superior separability.** We visualize the distribution of applicable and non-applicable inputs in Figure 6b. On the y-axis (PC1 of base hidden states), we observe that the two distributions are heavily entangled, indicating that standard dimensionality reduction techniques fail to provide a distinguishable boundary for identifying intervention targets. In contrast, the x-axis (Energy score $E(x)$) demonstrates a clear separation where two distributions exhibit minimal overlap. This observation empirically justifies the rationality of employing energy as a discriminative criterion for intervention applicability.

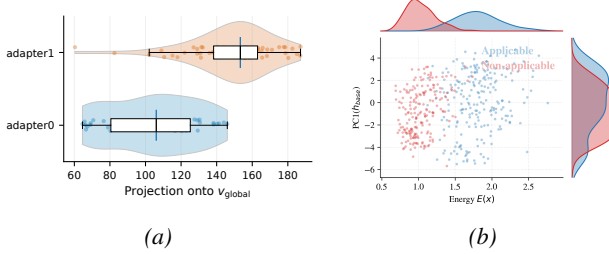

*(a)*        *(b)*

*Figure 6.* **Energy-based gating diagnostics.** (a) We project each adapter's intervention update $\Delta h$ onto a shared global direction $v_{\text{global}}$ (the normalized mean update across adapters) and plot the resulting projection distributions. Differences across adapters indicate complementary intervention behaviors rather than a single shared direction. (b) We plot **applicable** and **non-applicable** inputs. The y-axis shows $\text{PC1}(h_{\text{base}})$, where the two distributions are heavily entangled. In contrast, the Energy score $E(x)$ on the x-axis demonstrates clear separability with minimal overlap.

**Precise intervention control.** We further investigate the impact of MARI on internal representations by visualizing the shift before and after intervention in Figure 7. For **applicable** inputs (left), we observe a substantial distributional shift, quantified by a normalized mean shift $\|\Delta\mu\|/\sigma$ of 119.76. This indicates that the intervention effectively reorients the latent states to correct errors. In stark contrast, **non-applicable** inputs (right) remain virtually unchanged, exhibiting a negligible shift of only 10.22. This confirms that our energy-gating mechanism functions as an effective safety filter, permitting necessary corrections while shielding benign queries from unnecessary perturbation.

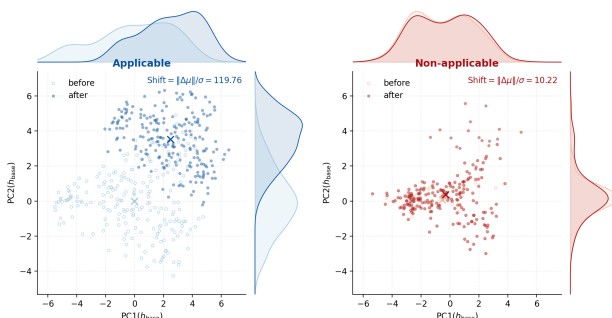

*Figure 7.* **Representation shift under intervention.** **Applicable** inputs exhibit a substantial mean shift, confirming effective intervention, whereas **non-applicable** inputs remain nearly unchanged, demonstrating MARI effectively shields benign queries.

**Inference Efficiency.** We evaluate computational overhead under the EasySteer benchmarking protocol (Xu et al., 2025), reporting First Token Latency (FTL), Tokens Per Second (TPS), and Total Latency (TTLT). Our routing uses *vectorized parallel scoring*, batching all experts in a single forward pass to compute $u_k(x)$, so the overhead is limited to a short prompt-only step and does not scale with $K$, keeping TTLT close to ReFT in practice. Further cost decomposition

and an illustrative example are given in Appendix B.4.

**Probe-stage memory overhead.** The energy score is computed in a short prompt-only probe stage before generation. This probe does not introduce an additional decoding or generation pass. Instead, the temporary probe activations are used only to aggregate propagation responses and determine the gate and routing decision, after which they are immediately discarded. Therefore, the additional activation memory is transient and scales approximately as $O(L_{\text{post}}Td)$, where $L_{\text{post}}$ is the number of post-injection layers, $T$ is the prompt length, and $d$ is the hidden dimension. Once the gate and router decisions are made, downstream decoding is executed only once, either with the frozen base model alone or with the selected adapter.

*Table 4.* **Inference efficiency comparison.** Results are reported as $\text{mean}_{\pm\text{std}}$ over 3 runs.

| Method | FTL (ms) ↓ | TPS (tok/s) ↑ | TTLT (s) ↓ |
|---|---|---|---|
| Base Model | $22.12_{\pm 0.15}$ | $476.26_{\pm 2.45}$ | $0.0223_{\pm 0.0005}$ |
| ITI | $22.14_{\pm 0.18}$ | $475.74_{\pm 3.12}$ | $0.0223_{\pm 0.0006}$ |
| NLITI | $23.78_{\pm 0.21}$ | $473.60_{\pm 4.05}$ | $0.0234_{\pm 0.0008}$ |
| ReFT | $43.64_{\pm 0.45}$ | $445.01_{\pm 5.23}$ | $0.0439_{\pm 0.0011}$ |
| **MARI** | $\mathbf{43.53}_{\pm 0.42}$ | $\mathbf{446.32}_{\pm 4.98}$ | $\mathbf{0.0448}_{\pm 0.0021}$ |

## 7. Conclusion

We introduced **MARI**, a parameter-efficient framework for precise, input-adaptive intervention. Our Competitive Multi-Adapter mechanism tailors the optimal steering direction and strength for each sample, resolving the heterogeneity of intervention requirements. Moreover, the Energy-Based Gating module leverages propagation dynamics to robustly filter benign inputs. Experiments confirm that MARI achieves state-of-the-art alignment performance across diverse model scales while preserving general capabilities.

**Limitations and future work.** While our study reveals nonlinear intervention requirements at a specific injection layer. A more holistic view of representation dynamics across layers, token positions, and generation time may further clarify how intervention effects propagate and evolve as trajectories through the model. We leave the development of trajectory-level analysis and multi-layer energy control as promising directions for future work.

## Acknowledgements

This work is partially supported by the University of Queensland School of Electrical Engineering and Computer Science (NS-2401), the Australian Research Council Discovery Project (DP230101753), the MBZUAI Research Fund (BF0100), JST PRESTO (JPMJPR23P5), JST CREST (JP-MJCR21M2), and JST NEXUS (JPMJNX25C4).

## Impact Statement

This paper presents work whose goal is to advance the field of Machine Learning. There are many potential societal consequences of our work, none of which we feel must be specifically highlighted here.

However, we believe it is important to contextualize the broader implications of our proposed framework, MARI, particularly in the domain of Large Language Model (LLM) alignment. The primary societal contribution of this work is the development of a robust mechanism to enhance the truthfulness, fairness, and safety of LLMs without compromising their general reasoning capabilities. By enabling precise, input-adaptive interventions, our method addresses a critical barrier in deploying LLMs known as the "alignment tax," where safety measures often degrade model utility. Specifically, by significantly improving performance on benchmarks like TruthfulQA and BBQ, MARI helps reduce the propagation of misinformation and social stereotypes, fostering more trustworthy and equitable AI systems. Furthermore, the energy-based gating mechanism ensures that interventions are applied only when necessary, providing a reliable safety filter for real-world applications ranging from customer service to educational tools.

Like many alignment techniques, representation intervention carries a potential dual-use risk. While our goal is to steer models toward desired behaviors, the underlying mechanisms of manipulating internal representations could theoretically be repurposed by malicious actors to suppress safety filters or induce harmful behaviors. However, we argue that understanding these internal control mechanisms is a prerequisite for effective defense. By exposing the geometric nature of alignment failures and providing a method to detect them via propagation energy, our work contributes to the interpretability and robustness of AI systems, ultimately aiding the community in building stronger defenses against adversarial manipulation.

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

# A. Experimental Settings Details

## A.1. Datasets and Evaluation Protocols

In this section, we provide detailed specifications for the benchmarks and metrics used in our evaluation.

**TruthfulQA (Lin et al., 2022).** This benchmark consists of 817 questions spanning 38 categories (e.g., health, law, finance).

- **Multiple-Choice (MC1 & MC2):** For MC1, we compute the log-likelihood of each answer choice given the question. The score is 1 if the best-scoring option is a true answer, and 0 otherwise. For MC2, we compute the total normalized probability mass assigned to the set of true answers.

- **Generation (BLEU & BLEURT):** We use the standard prompt: `"Q: <Question>`
  `nA:"` and generate greedy responses (temperature=0). Following standard practice, we evaluate semantic truthfulness by comparing the generated answer against the reference "correct answers" provided in the dataset using BLEU and a finetuned BLEURT checkpoint.

**BBQ (Bias Benchmark for QA) (Parrish et al., 2022).** We utilize the *disambiguated* subset of BBQ, where the context provides sufficient information to answer correctly without relying on stereotypes.

- **Metric:** We report Accuracy. A model is considered correct only if it selects the target non-stereotypical answer. Selecting the stereotype-consistent option or an "unknown" option is counted as a failure. We evaluate across all 9 social dimensions (e.g., Age, Gender, SES).

**Safety Refusal (Sorry-Bench) (Xie et al., 2025).** We employ a dataset of 450 harmful instructions covering diverse categories (e.g., Hate Speech, Dangerous Content).

- **Metric (Refusal Rate):** We employ a strictly prompt-based judge using MISTRAL-7B-INSTRUCT-V0.2. The judge classifies the model's response into "Refusal" (Safe) or "Compliance" (Unsafe). The score is the percentage of successful refusals.

**General Capabilities (MMLU, ARC).**

- **MMLU (Hendrycks et al., 2020):** We use the standard 5-shot formatting. We evaluate on the validation set for hyperparameter tuning and report results on the test set.

- **ARC (Clark et al., 2018):** We use the 25-shot setting for both ARC-Easy and ARC-Challenge to robustly estimate reasoning capabilities.

## A.2. Baselines

We compare our proposed MARI framework against the following state-of-the-art representation intervention paradigms:

- **Vanilla**: The original pre-trained Large Language Model (LLM) without any intervention or modification during inference. This serves as the lower bound to assess the raw capabilities of the model.

- **CAA (Contrastive Activation Addition)**: A static steering method that computes a global intervention vector by averaging the differences in activations between positive (e.g., truthful) and negative (e.g., hallucinated) example pairs. This fixed vector is added to the hidden states of all inputs during inference with a constant coefficient.

- **ITI (Inference-Time Intervention)**: A technique that identifies a sparse set of attention heads that possess high linear probing accuracy for a target attribute. During inference, it shifts the activations of these specific heads along the "truthful" direction derived from the linear probe, scaled by their standard deviation.

- **NL-ITI (Non-Linear Inference-Time Intervention)**: An extension of the ITI framework that replaces the linear probe with non-linear classifiers. It aims to capture more complex, non-linear geometries in the activation space to determine the intervention direction, theoretically offering more precise control than linear methods.

- **ReFT (Representation Finetuning)**: A parameter-efficient intervention method that learns a low-rank update matrix (similar to LoRA) directly on the hidden states. Unlike static steering vectors, ReFT optimizes these intervention parameters on a small set of labeled data to manipulate representations toward the target behavior.

### .1. Experimental Setup

**Hardware.** All experiments were conducted on NVIDIA RTX PRO 6000 Blackwell GPUs.

**Training data budget and construction.** We follow a data-efficient but fully specified protocol for constructing the training pool used for subspace estimation and adapter training. For TruthfulQA, the dataset contains 817 questions in total, and we use a training pool of 200 questions sampled uniformly at random. For all other datasets, we use a fixed fraction of the dataset total, set to 20%, to form the training pool. This fraction is applied consistently across models, including 32B models, to ensure comparable data budgets and stable estimation.

**PCA subspace estimation.** We construct the PCA set $D_{\text{pca}}$ using half of the training pool, and we use the question text only. We extract hidden states from these questions and estimate the in-field basis $B$ with PCA rank $k = 16$.

**Probe and adapter training.** We use the remaining half of the training pool as $D_{\text{train}}$. The probe $g_\phi$ and all adapters are trained on the same set $D_{\text{train}}$ to avoid introducing additional data sources. Unless otherwise stated, we use probe rank $r_{\text{probe}} = 2$.

**Hyperparameter sweeps and reporting.** We sweep the number of adapters $K \in \{1, 2, 3, 4, 5, 6\}$ and the gating parameter $\rho \in \{0.7, 0.8, 0.9\}$, which controls the energy threshold $\tau_E$ via a fixed quantile rule. For each configuration, we report the mean and standard deviation over three runs with different random seeds.

**Energy gate calibration.** We calibrate $\tau_E$ once using a small control set $D_{\text{ctrl}}$ and keep it unchanged for all reported benchmarks. $D_{\text{ctrl}}$ is constructed from two sources: held-out examples from the intervention-attribute datasets that are disjoint from all reported test sets, and a small subset of ARC-Easy training questions. Each control example is labeled as applicable or non-applicable based on its computable intervention outcome on $D_{\text{ctrl}}$. We then set $\tau_E$ as the $\rho$ quantile of the energy distribution on the non-applicable subset of $D_{\text{ctrl}}$. All components used for $D_{\text{ctrl}}$ are disjoint from $D_{\text{pca}}$ and $D_{\text{train}}$.

## A. Deferred equations for Sections 4.2 and 4.3

**Per-adapter losses (used in Section 4.2).** For multiple-choice tasks, let $\mathcal{Y}(x) = \{y^{(1)}, \ldots, y^{(M)}\}$ be the candidate set and let $z_k(x) \in \mathbb{R}^M$ be the option scores under adapter $k$ (e.g., sequence log-likelihoods). We set

$$\ell_k(x, y) := \text{CE}(z_k(x), y). \tag{20}$$

For free-form generation with a target sequence $y = (y_1, \ldots, y_T)$, define the teacher-forced next-token distribution $p_{k,t}(\cdot) := p_{\theta, \psi_k}(\cdot \mid x, y_{<t})$. We set

$$\ell_k(x, y) := -\sum_{t=1}^{T} \log p_{k,t}(y_t). \tag{21}$$

**Usage balancing (referenced in Section 4.2).** Within a minibatch $\mathcal{B}$, we form a differentiable soft assignment from current losses:

$$Z(x, y) := \sum_{j=1}^{K} \exp(-\ell_j(x, y)/T_{\text{bal}}),$$

$$q_k(x, y) := \exp(-\ell_k(x, y)/T_{\text{bal}}) / Z(x, y), \tag{22}$$

$$p_k := \frac{1}{|\mathcal{B}|} \sum_{(x,y) \in \mathcal{B}} q_k(x, y).$$

We penalize deviation from uniform usage:

$$\mathcal{L}_{\text{bal}} := \sum_{k=1}^{K} \left(p_k - \frac{1}{K}\right)^2. \tag{23}$$

**Directional / subspace diversity (referenced in Section 4.2).** *(i) Output-subspace orthogonality.* Let $Q_k = \text{orth}(U_k) \in \mathbb{R}^{d \times r}$ be an orthonormal basis of $\text{span}(U_k)$ obtained via a reduced QR decomposition. We penalize inter-adapter overlap by

$$\mathcal{L}_{\text{inter}} := \frac{2}{K(K-1)} \sum_{1 \le i < j \le K} \left\| Q_i^\top Q_j \right\|_F^2. \tag{24}$$

*(ii) Per-sample direction diversity.* For the injection-site hidden state $h(x)$, define the (scaled) additive update produced by adapter $k$ as

$$\delta_k(x) := \gamma \, s_k \, \Delta_{\psi_k}\big(h(x)\big) \in \mathbb{R}^d, \tag{25}$$

and its normalized direction $\widehat{\delta}_k(x) = \delta_k(x)/\|\delta_k(x)\|_2$. We penalize pairwise squared cosine similarity:

$$\mathcal{L}_{\text{out}} := \mathbb{E}_{(x,y)} \left[ \frac{2}{K(K-1)} \sum_{1 \le i < j \le K} \big\langle \widehat{\delta}_i(x), \widehat{\delta}_j(x) \big\rangle^2 \right]. \tag{26}$$

**Entropy uncertainty definitions (referenced in Section 4.2).** For **multiple-choice**, define $p_k(\cdot \mid x) = \text{softmax}(z_k(x))$ and set

$$u_k(x) := H\big(p_k(\cdot \mid x)\big). \tag{27}$$

For **free-form generation**, let $p_{k,t}(\cdot) := p_{\theta, \psi_k}(\cdot \mid x, \hat{y}_{<t}^{(k)})$ be the next-token distribution at step $t$ under adapter $k$, where $\hat{y}^{(k)}$ denotes the greedy prefix generated under that adapter. We average next-token entropy over the first $T_{\text{ent}}$ greedy decoding steps:

$$u_k(x) := \frac{1}{T_{\text{ent}}} \sum_{t=1}^{T_{\text{ent}}} H\big(p_{k,t}(\cdot)\big). \tag{28}$$

**Propagation response and projection operators (referenced in Section 4.3).** We define the per-layer propagation response (used by $E(\cdot)$ in Eq. (12)) as

$$e_m(x; \alpha) \triangleq \big\| h_{p^\star}^{(\alpha, m)}(x) - h_{p^\star}^{(m)}(x) \big\|_2, \qquad m = l^\star, \dots, L. \tag{29}$$

For the off-subspace penalty in Eq. (15), we use a rank-$k$ PCA basis $B \in \mathbb{R}^{d \times k}$ (columns orthonormal) and define

$$\Pi_B(v) := BB^\top v, \qquad \Pi_B^\perp(v) := \big(I - BB^\top\big)v. \tag{30}$$

## B. Theoretical Analysis Details

### B.1. Entropy routing: proof of Theorem 5.1

**Definitions.** Let $\ell_k(x, y) \in [0, L]$ be the supervised loss for expert $k$. Define the oracle and entropy-selected experts as

$$k^\star(x, y) := \arg\min_k \ell_k(x, y), \qquad \hat{k}(x) := \arg\min_k u_k(x),$$

where $u_k(x)$ is the entropy-based uncertainty score used in the main paper. Define

$$R_{\text{ent}} := \mathbb{E}\big[\ell_{\hat{k}(x)}(x, y)\big], \quad R_{\text{min}} := \mathbb{E}\big[\min_k \ell_k(x, y)\big] = \mathbb{E}\big[\ell_{k^\star(x,y)}(x, y)\big], \quad R_{\text{single}} := \min_j \mathbb{E}\big[\ell_j(x, y)\big].$$

Let the misrouting event be $\mathcal{M} := \{\hat{k}(x) \ne k^\star(x, y)\}$ and $\eta := \Pr(\mathcal{M})$.

**Proof.** We start from

$$\begin{aligned}
R_{\text{ent}} - R_{\text{min}} &= \mathbb{E}\Big[\ell_{\hat{k}(x)}(x, y) - \ell_{k^\star(x,y)}(x, y)\Big] \\
&= \mathbb{E}\Big[\big(\ell_{\hat{k}(x)}(x, y) - \ell_{k^\star(x,y)}(x, y)\big) \mathbf{1}\{\mathcal{M}\}\Big],
\end{aligned} \tag{31}$$

since the difference is zero whenever $\hat{k}(x) = k^\star(x, y)$. Because $\ell_k(x, y) \in [0, L]$ for all $k$, we have

$$0 \le \ell_{\hat{k}(x)}(x, y) - \ell_{k^\star(x,y)}(x, y) \le L \quad \text{on } \mathcal{M}.$$

Plugging this into Eq. (31) yields

$$R_{\text{ent}} - R_{\text{min}} \le L \Pr(\mathcal{M}) = L \eta, \tag{32}$$

i.e.,

$$R_{\text{ent}} \le R_{\text{min}} + L \eta. \tag{33}$$

Finally, rewrite $R_{\text{min}} = R_{\text{single}} - (R_{\text{single}} - R_{\text{min}})$ to obtain

$$R_{\text{ent}} \le R_{\text{single}} - (R_{\text{single}} - R_{\text{min}}) + L \eta.$$

If $R_{\text{single}} - R_{\text{min}} > L \eta$, then the right-hand side is strictly smaller than $R_{\text{single}}$, hence $R_{\text{ent}} < R_{\text{single}}$. $\qquad\square$

### B.2. Empirical lower bound for entropy-to-correctness under shared-scale competition

**Setup (post-hoc diagnostic).** To operationalize the misrouting term in Theorem 5.1 without introducing additional tuning, we run a post-hoc cross-validation diagnostic on the **test sets** of TruthfulQA (MC1/MC2) and BBQ using **Llama-3-8B-Instruct**. This diagnostic is used *only* to quantify a lower bound on routing reliability and is not used to select hyperparameters, set thresholds, or report final benchmark scores.

**Shared-scale competitive loss and entropy score.** During training, experts are optimized under a competitive objective on a shared loss scale. At inference, each expert $k \in [K]$ produces an entropy-based uncertainty score $u_k(x)$ (Eq. (10)). For each example $(x, y)$, define the single-expert correctness indicator

$$c_k(x, y) = \mathbf{1}[\hat{y}_k(x) = y], \tag{34}$$

where $\hat{y}_k(x)$ is the prediction produced when selecting expert $k$. Because experts are trained with a shared-scale competitive loss, the entropy scores are directly comparable across experts, enabling a meaningful "entropy-to-correctness" assessment.

**Cross-validation protocol.** We perform $F$-fold cross-validation over each test set. In each fold, we compute $(u_k(x), c_k(x, y))$ for all experts and all examples in the held-out fold, and evaluate whether the entropy-based selection agrees with the best expert under correctness. Concretely, define

$$\hat{k}(x) = \arg\min_k u_k(x), \qquad k^\star(x, y) = \arg\max_k c_k(x, y), \tag{35}$$

and the fold-wise agreement accuracy

$$\text{Acc}_{\text{agree}} = \frac{1}{|D|} \sum_{(x,y) \in D} \mathbf{1}\left[\hat{k}(x) = k^\star(x, y)\right]. \tag{36}$$

We report the minimum agreement across folds as a conservative lower bound:

$$\text{LB}_{\text{agree}} = \min_{f \in [F]} \text{Acc}_{\text{agree}}^{(f)}. \tag{37}$$

**Results (lower bound).** On TruthfulQA-MC1, TruthfulQA-MC2, and BBQ test sets for Llama-3-8B-Instruct, we consistently observe a non-trivial lower bound

$$\text{LB}_{\text{agree}} \ge 81.74\%, \tag{38}$$

indicating that the entropy-based criterion selects a correctness-maximizing expert with high reliability under this shared-scale competitive setting. This empirical lower bound supports the claim that $\eta$ remains small in practice for the reported configurations.

## B.3. Energy gating: geometric bound and proof

**Energy score and first-order sensitivity.** Recall the per-layer propagation response

$$e_m(x;\alpha) := \left\| h_{p^\star}^{(\alpha,m)}(x) - h_{p^\star}^{(m)}(x) \right\|_2, \qquad m \in \{l^\star, \dots, L\},$$

and the energy score

$$E(x;\alpha) := \text{median}\left( \{e_m(x;\alpha)\}_{m=l^\star}^L \right).$$

For analysis, define $\bar{E}(x;\alpha) := \max_m e_m(x;\alpha)$, so that $E(x;\alpha) \leq \bar{E}(x;\alpha)$, which states that for each $m$ and sufficiently small $\alpha$,

$$h_{p^\star}^{(\alpha,m)}(x) - h_{p^\star}^{(m)}(x) = \alpha J_m(x)\delta_\phi(x) + o(\alpha), \qquad e_m(x;\alpha) = \alpha\|J_m(x)\delta_\phi(x)\|_2 + o(\alpha), \tag{39}$$

where $J_m(x) \in \mathbb{R}^{d \times d}$ is the Jacobian of the mapping from the injected representation to $h_{p^\star}^{(m)}$ at input $x$.

**In-field subspace and probe alignment.** Let $B \in \mathbb{R}^{d \times k}$ denote the in-field subspace at the injection layer, with orthogonal projectors $\Pi_B$ and $\Pi_B^\perp$. Assume the probe update $\delta_\phi(x)$ satisfies the alignment bounds on the domain of interest: there exist constants $S > 0$ and $\varepsilon \geq 0$ such that

$$\|\Pi_B \delta_\phi(x)\|_2 \leq S, \qquad \|\Pi_B^\perp \delta_\phi(x)\|_2 \leq \varepsilon. \tag{40}$$

**Propagation gains.** Define the (layerwise) restricted operator norms

$$\kappa_B(x) := \max_{m \in \{l^\star, \dots, L\}} \|J_m(x)\Pi_B\|_{2 \to 2}, \qquad \Gamma(x) := \max_{m \in \{l^\star, \dots, L\}} \|J_m(x)\Pi_B^\perp\|_{2 \to 2}. \tag{41}$$

Here $\kappa_B(x)$ captures propagation gain along in-field directions, while $\Gamma(x)$ captures propagation gain from $B^\perp$ components.

**Proof of Theorem 5.2.** Fix $x$ and $m \in \{l^\star, \dots, L\}$. Decompose the probe direction as $\delta_\phi(x) = \Pi_B \delta_\phi(x) + \Pi_B^\perp \delta_\phi(x)$. Using Eq. (39) and the triangle inequality,

$$\begin{aligned} e_m(x;\alpha) &= \alpha\|J_m(x)\delta_\phi(x)\|_2 + o(\alpha) \\ &\leq \alpha\left( \|J_m(x)\Pi_B\delta_\phi(x)\|_2 + \|J_m(x)\Pi_B^\perp\delta_\phi(x)\|_2 \right) + o(\alpha) \\ &\leq \alpha\left( \|J_m(x)\Pi_B\|_{2 \to 2}\|\Pi_B\delta_\phi(x)\|_2 + \|J_m(x)\Pi_B^\perp\|_{2 \to 2}\|\Pi_B^\perp\delta_\phi(x)\|_2 \right) + o(\alpha). \end{aligned} \tag{42}$$

Taking the maximum over $m$ and applying Eq. (41) yields

$$\bar{E}(x;\alpha) \leq \alpha\left( \kappa_B(x)\|\Pi_B\delta_\phi(x)\|_2 + \Gamma(x)\|\Pi_B^\perp\delta_\phi(x)\|_2 \right) + o(\alpha). \tag{43}$$

Using the alignment bounds in Eq. (40) gives

$$\bar{E}(x;\alpha) \leq \alpha\left( \kappa_B(x) S + \Gamma(x) \varepsilon \right) + o(\alpha). \tag{44}$$

Since $E(x;\alpha) \leq \bar{E}(x;\alpha)$, the same bound holds for $E(x;\alpha)$.

For non-applicable inputs $x \sim \mathcal{D}_{\text{non}}$, if $\kappa_B(x) \leq \kappa_{\text{non}}$ (attenuated in-field propagation) and the off-subspace penalty yields small $\varepsilon$, then $E(x;\alpha)$ is small in the small-$\alpha$ regime, which is exactly the statement in Eq. (19). $\qquad \square$

## B.4. Inference Cost Analysis

**Cost decomposition.** We decompose inference cost into (i) base-model forward compute, (ii) router evaluation, and (iii) intervention injection:

$$\text{Cost} \approx F \cdot (\text{forward tokens}) + F \cdot (\text{router tokens}) + A \cdot (\text{\# injected token-layers}),$$

where $F$ is the per-token base-model forward cost and $A$ is the per-token-layer injection cost. In typical transformer settings, $A \ll F$ (e.g., injection $O(Hr)$ vs. base layer $O(H^2)$), so injection FLOPs are negligible compared to extra base-model forwards.

**Reference: single-edit (always-on, single injection).** Many single-edit baselines apply one injection per query without any additional probe or routing passes. For candidate-based scoring,

$$\text{Cost}_{\text{single}} \approx F(P + CL) + A S_I L_I,$$

where $P$ is prompt length, $C$ is the number of candidates, $L$ is the average candidate length, and $S_I, L_I$ are the number of injected tokens and layers (for our single-token/single-layer setting, $S_I L_I = 1$).

**Router evaluation cost (entropy routing).** Entropy routing in Eq.(10) selects $\hat{k}(x) = \arg\min_k u_k(x)$, which requires evaluating the uncertainty scores $\{u_k(x)\}_{k=1}^K$. Let $T_{\text{route}}$ denote the token budget used to compute $u_k(x)$ (*multiple-choice:* $T_{\text{route}} = CL$ via option-score entropy; *generation:* $T_{\text{route}} = T_{\text{ent}}$ via the first $T_{\text{ent}}$ next-token entropies). In a naive *sequential* implementation, computing $\{u_k(x)\}$ incurs an additional

$$\text{Cost}_{\text{route}} \approx (K - 1) F(P + T_{\text{route}}),$$

since the selected expert's forward can be reused for answering.

*Implementation note (latency vs. FLOPs).* While the total FLOPs of routing scale linearly with $K$, our wall-clock latency measurements (Table 3) use a batched implementation: we stack the $K$ expert variants along the batch dimension and (when applicable) reuse the prompt KV cache. For the small $K$ used in our experiments ($K \leq 6$), this parallelism makes the latency increase sublinear in $K$ even though the arithmetic FLOPs increase.

**Ours (energy gate + router + single actuation).** The energy probe performs $m$ extra *prompt-only* forwards (in our experiments $m \leq 3$). Moreover, when the energy gate predicts *non-applicable* inputs (i.e., $\alpha(x) = \alpha_{\text{safe}} = 0$), we directly fall back to the frozen base model and *skip* router evaluation. Let $q := \Pr[E(x; \alpha_{\text{probe}}) \geq \tau_E]$ be the fraction of queries on which the gate activates actuation. Then the expected cost is

$$\mathbb{E}\big[\text{Cost}_{\text{ours}}\big] \approx F(P + CL) + mFP + q(K - 1)F(P + T_{\text{route}}) + q A S_I L_I.$$

Thus, relative to a single-edit baseline,

$$\frac{\mathbb{E}[\text{Cost}_{\text{ours}}]}{\text{Cost}_{\text{single}}} \approx 1 + \frac{mP}{P + CL} + q(K - 1)\frac{P + T_{\text{route}}}{P + CL},$$

where we again use $A \ll F$. Importantly, $K$ does *not* multiply the *final generation* cost: only one adapter is applied for actuation, and the injection term remains $O(1)$ in $K$; the $K$-dependence arises only in the router evaluation term.

**Reference: ReFT (single adapter, multi-token/multi-layer injection).** ReFT uses a single adapter but injects across multiple tokens and layers:
$$\text{Cost}_{\text{ReFT}} \approx F(P + CL) + A S_R L_R,$$

where $S_R, L_R$ are the number of injected tokens and layers.

**Ratio to ReFT.**

$$\frac{\mathbb{E}[\text{Cost}_{\text{ours}}]}{\text{Cost}_{\text{ReFT}}} \approx \frac{F(P + CL) + mFP + q(K - 1)F(P + T_{\text{route}}) + q A S_I L_I}{F(P + CL) + A S_R L_R} \approx 1 + \frac{mP}{P + CL} + q(K - 1)\frac{P + T_{\text{route}}}{P + CL},$$

again using $A \ll F$.

## C. Related Works

**Geometry of Representations.** The Linear Representation Hypothesis serves as the foundational premise for most existing representation intervention methods. This hypothesis posits that neural networks encode semantic concepts as linear directions within their activation space (Mikolov et al., 2013; Bolukbasi et al., 2016; Elhage et al., 2021; Park et al., 2023). Under this assumption, approaches such as Activation Steering reduce the manipulation of a specific attribute to identifying

and shifting a single vector corresponding to that concept (Turner et al., 2024; Zou et al., 2023; Wu et al., 2024). However, recent studies employing advanced interpretability tools have revealed that certain attributes exhibit non-linear geometric structures (Cunningham et al., 2023). For instance, representations of temporal concepts have been shown to form circular structures to capture cyclical patterns. Similarly, models construct internal manifolds reflecting the structural relationships of physical space or color (Abdou et al., 2021; Patel & Pavlick, 2022; Gurnee & Tegmark, 2024). Consequently, intervention methods predicated solely on the linear representation hypothesis may be insufficient to precisely intervene on target attributes without disrupting other untargeted properties.

