# OpenReview forum: "Multi-Adapter Representation Interventions via Energy Calibration"
_ICML.cc/2026/Conference — ICML 2026 regular_

### Official Review · Reviewer_NfKN · 2026-03-06

**Soundness:** 3
**Presentation:** 3
**Significance:** 3
**Originality:** 3
**Overall Recommendation:** 4
**Confidence:** 4

**Summary:**

This paper proposes MARI, a test-time low-rank activation intervention method for aligning LLMs without updating backbone weights. It trains multiple competing adapters and uses entropy-based routing to select one per input, while an energy-calibrated gate decides whether to intervene or fall back to the base model, aiming to improve alignment metrics with minimal capability loss. Experiments show that MARI’s improve target alignment metrics while reducing over-intervention and preserving general performance better than single-adapter or always-on baselines.

**Compliance With Llm Reviewing Policy:**

Affirmed.

**Final Justification:**

My concerns have been addressed. I remain supportive of the paper’s acceptance

**Key Questions For Authors:**

Please refer to the weakness.

**Limitations:**

Yes

**Strengths And Weaknesses:**

**Strengths**:

1.	The proposed approach is motivated by a well-justified premise, supported by a clear theoretical analysis.

2.	The empirical evaluation is fairly extensive, covering a range of state-of-the-art models and multiple benchmark datasets.

3.	Using energy propagation to assess intervention sensitivity is a relatively novel idea.

**Weaknesses**:

1.	The paper appears to build on fine-tuning–based representation editing, which would plausibly yield stronger performance than non–fine-tuning approaches such as ITI. Therefore, the authors should include additional comparisons against fine-tuning–based representation editing methods, e.g., LoFit [1].

2.	Given that the method requires fine-tuning, it is unclear whether it remains robust in extremely low-data regimes (e.g., clinical diagnosis scenarios, only with 10 or fewer samples).

3.	Because the approach trains multiple auxiliary modules (including experts, probes), the overall training cost may increase. A more thorough analysis of training efficiency would strengthen the paper.

4.	The paper noted in line 160 that task-specific losses differ across tasks. This raises the question of whether the method’s cross-task generalization is limited.

5.	In my experience, the choice of the edited layer can substantially affect performance. Do the authors provide any empirical guidance or ablation on layer selection, and how should practitioners identify the optimal layer in practice?

[1]Yin F, Ye X, Durrett G. Lofit: Localized fine-tuning on llm representations[J]. Advances in neural information processing systems, 2024, 37: 9474-9506.

---

> ### Author Rebuttal · Authors · 2026-03-30
>
> # Response to Reviewer NfKN
>
> We thank the reviewer for recognizing the paper's clear motivation, solid theory, broad evaluation scope, and the novelty of the propagation-energy design.
>
> ### W1: Comparison with LoFiT
>
> **A1:** We additionally compared MARI against **LoFiT**, a representative fine-tuning-based representation editing method:
>
> | Method  |   MC1 |   MC2 |   BBQ | MMLU | ARC-E | ARC-C |
> | ------- | ----: | ----: | ----: | ---: | ----: | ----: |
> | Vanilla | 28.70 | 45.03 | 0.608 | 65.9 |  77.3 |  51.4 |
> | LoFiT   | 61.84 | 66.16 | 0.754 | 64.5 |  77.0 |  51.4 |
> | EG-MARI | 64.81 | 67.39 | 0.792 | 66.6 |  78.0 |  52.1 |
>
> EG-MARI consistently outperforms LoFiT on the reported metrics in this matched comparison.
>
> ### W2: Extremely Low-Data Regime
>
> **A2:** To directly test this, we ran a small-data sensitivity pilot on **TruthfulQA**, reducing the training pool size while keeping the rest of the training and evaluation protocol fixed. The resulting routed MC1 accuracies are:
>
> | # training examples |    MC1 |
> | ------------------- | -----: |
> | Baseline            | 0.2870 |
> | 10                  | 0.3380 |
> | 20                  | 0.3472 |
> | 50                  | 0.4398 |
> | 100                 | 0.5093 |
>
> On this TruthfulQA pilot, MARI already improves over the baseline even in the 10-sample regime, and performance increases as the number of training examples grows.
>
> ### W3: Training Efficiency
>
> **A3:** MARI trains multiple lightweight modules (experts plus a small probe/calibrator), while the entropy router itself is parameter-free, but its training cost is **not** equivalent to training multiple independent adapters in separate runs. MARI uses a joint competitive objective: the backbone-side computation is shared within each training step, and the experts are optimized together in a single training run. As a result, the additional cost is a modest constant-factor overhead, rather than a multiplicative \(K\times\) increase.
>
> To make this concrete, we measured the end-to-end training cost on Llama-3-8B-Instruct under the same hardware setup:
>
> | Method                            |        Trainable Params | Total Wall-clock | GPU-hours | Peak GPU Memory |
> | --------------------------------- | ----------------------: | ---------------: | --------: | --------------: |
> | MARI (compete, \(K=3\)) |                  66,948 |           ~9m48s |    ~0.163 |        17.4 GiB |
> | LoFiT                             | 135,168 optimized total |           18m26s |     0.307 |       33.3 GiB |
>
> These results show that MARI remains efficient in practice: despite using multiple experts and a lightweight routing component, joint competitive training keeps both wall-clock cost and memory usage well below LoFiT.
>
> ### W4: Cross-Task Generalization
>
> **A4:** Our claim is **not** that one trained editor transfers zero-shot across heterogeneous tasks. Rather, MARI is a **task-agnostic intervention framework** whose core mechanism remains the same across tasks: the same competitive multi-adapter routing and energy-gating design is used throughout, while the supervised loss is instantiated according to the output format, e.g., cross-entropy for multiple-choice tasks and teacher-forced negative log-likelihood for generation tasks.
>
> MARI is designed for cross-task applicability, not for using one identical loss across all task formats. Empirically, the same framework is effective across TruthfulQA, BBQ, and Sorry-Bench despite their different supervision forms and data characteristics.
>
>
>
> ### W5: Guidance on Layer Selection
>
> **A5:** We agree that the edited layer is an important hyperparameter. Our practical selection procedure is as follows. In our experiments, the injection layer was selected by a lightweight validation sweep using a **single-adapter (`K=1`)** variant, and the selected layer was then kept fixed for the full MARI model. This keeps the search cost modest and avoids a much more expensive joint search over both layer and multi-adapter settings.
>
> For example, for Llama-3-8B (**32 layers**), we searched over the middle-to-late layers, specifically **layers 16 to 28**, and selected the best-performing one on the validation set.
>
> This choice is also consistent with prior empirical findings in the steering literature that intervention is often most effective in intermediate or middle-to-late layers; see, e.g., Wang et al., *Beyond Prompt Engineering: Robust Behavior Control in LLMs via Steering Target Atoms* (ACL 2025), and Zhao et al., *Steering Knowledge Selection Behaviours in LLMs via SAE-Based Representation Engineering* (NAACL 2025).

---

> > ### Author Rebuttal · Reviewer_NfKN · 2026-04-01
> >
> > I thank the authors for their thorough rebuttal and additional experiments, which have addressed my concerns. I remain supportive of the paper’s acceptance

---

### Official Review · Reviewer_PBZa · 2026-03-12

**Soundness:** 2
**Presentation:** 3
**Significance:** 2
**Originality:** 2
**Overall Recommendation:** 4
**Confidence:** 4

**Summary:**

The paper introduces MARI, a novel, inference-time alignment framework for LLMs. The authors observe that traditional representation interventions apply a static, uniform steering vector or matrix across all inputs, which fails to account for input heterogeneity and frequently degrades general model capabilities. To solve this, MARI proposes two core components: 1 a Competitive Multi-Adapter mechanism that trains multiple low-rank experts and routes inputs to the most suitable adapter using predictive entropy, and 2 an Energy-Based Gate that utilizes the internal propagation dynamics of a small probe update to decide whether an intervention should be applied or suppressed. The framework demonstrates strong performance improvements on TruthfulQA, BBQ, and safety benchmarks while preserving reasoning capabilities on MMLU and ARC.

**Compliance With Llm Reviewing Policy:**

Affirmed.

**Final Justification:**

All my concerns have been addressed.

**Key Questions For Authors:**

Could you please compare MARI with more recent works, including MSRS, MAT-Steer, PIXEL, and DIRECTER, to address the aforementioned weaknesses?

Jiang et al, MSRS: Adaptive Multi-Subspace Representation Steering for Attribute Alignment in Large Language Models, 2025

Nguyen et al, Multi-Attribute Steering of Language Models via Targeted Intervention, 2025

Yu et al, PIXEL: Adaptive Steering Via Position-wise Injection with eXact Estimated Levels under Subspace Calibration, 2025

Kang et al, Enhancing Instruction Following of LLMs via Activation Steering with Dynamic Rejection, 2026

**Limitations:**

No. Perhaps the authors can acknowledge:
1. the constraints of using a single, static injection layer compared to multi-layer or dynamic-layer steering
2. the potential bottlenecks of hard-routing when dealing with multi-attribute conflicts
3. the reliance on a well-calibrated control set for the energy threshold, which may require retuning for highly specialized downstream distributions.

**Strengths And Weaknesses:**

Strengths:
1. The Energy-Based Gate is a highly original, label-free mechanism for mitigating oversteering. Measuring intervention applicability via the Euclidean propagation response of a small probe in the latent space is computationally efficient and theoretically well-grounded.
2. MARI effectively solves the alignment tax problem that plagues static methods like ITI and CAA. By falling back to the base model when the energy threshold is not met, MARI actively preserves MMLU and ARC performance.
3. The integration of the multi-adapter routing allows MARI to achieve exceptional accuracy on alignment benchmarks, significantly outperforming standard static ReFT and additive steering methods.

Weaknesses:
1. MARI relies on an entropy-based hard routing strategy, meaning a single query is sent to a single discrete expert adapter. If an input requires simultaneous alignment across multiple axes (e.g., debiasing and hallucination correction), a single adapter may struggle. Contemporary works like MSRS  and MAT-Steer  solve this by allocating orthogonal subspaces or vectors that can be continuously blended, which is a structural advantage MARI lacks.
2. The framework intervenes at a single, fixed layer-position pair $(l^*, p^*)$. Recent literature, such as the PIXEL framework, demonstrates that the optimal intervention layer varies significantly depending on the specific input query, and utilizing dynamic inference-time position scanning can drastically reduce the required steering magnitude.
3. MARI's energy gate makes a binary decision: either apply the full adapter intervention or fall back entirely to the base model. This lacks the nuance of continuous modulation seen in frameworks like DIRECTER, which progressively reduces steering strength layer-by-layer based on output plausibility.

---

> ### Author Rebuttal · Authors · 2026-03-30
>
> # Response to Reviewer PBZa
> We thank the reviewer for the careful reading and for recognizing the originality of the energy gate and the strong preservation on MMLU/ARC. We agree that the paper should position MARI more clearly against adjacent adaptive-steering directions, and we address each point below.
> ### W1: Hard Routing and Blended Steering
> **A1:** We agree that simultaneous **multi-attribute** intervention is important, but our paper studies a different problem: **single-target intervention** with strong **input-wise heterogeneity**, where different inputs may need different correction patterns, and some should not be intervened on at all. In this setting, hard routing is a simple and effective design that encourages expert specialization while keeping the intervention path lightweight. To position MARI more concretely, we also compared it with MSRS and MAT-S in a matched setting:
>
> | Method | MC1 | MC2 | BBQ |
> |---|---:|---:|---:|
> | Baseline | 28.70 | 45.03 | 0.608 |
> | MAT-S| 30.31 | 49.85 | 0.632 |
> | MSRS | 35.31 | 55.32 | 0.665 |
> | **MARI** | **64.81** | **67.39** | **0.792** |
>
> These results show that, in the **single-target intervention** setting studied in this paper, MARI is highly competitive despite using hard routing rather than continuous blending. We will clarify this scope explicitly in the paper: MARI is designed for input-adaptive single-target intervention, not for full simultaneous multi-attribute composition. We will clearly state this scope limitation in the limitations section of the paper, and discuss methods such as MSRS and MAT-Steer as promising directions for future research on multi-attribute intervention.
> ### W2: Fixed Site vs. Dynamic Layer Selection
> **A2:** We appreciate the suggestion. First, the intervention site in MARI is not chosen arbitrarily; it is selected by validation hyperparameter search under the standard representation-intervention setup. We additionally evaluated a PIXEL-style adaptive site-selection variant. In our tested setting, this variant brought only marginal gains on TruthfulQA, while incurring a much more noticeable inference-time overhead:
>
> | Setting                               | TruthfulQA MC1 | TruthfulQA MC2 | TTLT (s) | Relative latency |
> | ------------------------------------- | -------------: | -------------: | -------: | ---------------: |
> | Fixed shared site (MARI)              |          64.81 |          67.39 |   0.0448 |            1.00x |
> | Adaptive site selection (PIXEL-style) |          65.23 |          67.88 |   0.0582 |            1.30x |
>
> These results suggest that adaptive site selection is a meaningful complementary extension, but in our setting, its additional benefit is modest relative to the added inference cost. In addition, the fixed-layer design keeps the contribution of routing and gating identifiable, rather than entangling them with gains from site search.
> ### W3: Binary Gate vs. Continuous Modulation
> We thank the reviewer for pointing us to DIRECTER; we note it appeared after our submission deadline but discuss it here as a relevant adjacent direction. The two methods address different control problems: DIRECTER modulates steering strength during decoding, while MARI decides whether and how to intervene per input. In our setting, the primary failure mode is unnecessary intervention on non-applicable inputs, making binary fallback a deliberate choice. To test this, we implemented a continuous-strength variant where intervention strength varies with the energy margin relative to the threshold:
>
> | Setting                     | TruthfulQA MC1 | TruthfulQA MC2 | MMLU |
> | --------------------------- | -------------: | -------------: | ---: |
> | Binary fallback (MARI)      |          64.81 |          67.39 | 66.6 |
> | Continuous-strength variant |          65.58 |          68.12 | 63.9 |
>
> The continuous variant provides only limited additional alignment gains, but preserves general capability substantially worse than binary fallback. This is consistent with the design goal of MARI: for inputs that should not be intervened on, even attenuated nonzero actuation can still introduce unnecessary perturbation. Therefore, we adopt binary fallback not because a continuous variant is impossible, but because it is the more effective design for preserving general capability in our setting.
> ### L1&2: Scope and Limitations
> **A4:** We agree that the paper’s scope and limitations should be stated more clearly. MARI focuses on **single-attribute intervention**—managing input-wise heterogeneous correction and applicability—rather than broader multi-attribute or adaptive steering. We will clarify this scope more explicitly in the limitations section.
> ### L3: Control set
> **A5:** `τ_E` is calibrated once on a held-out set (disjoint from MMLU/ARC-C) and remains fixed across all evaluations, demonstrating meaningful robust transfer. See R3WeA-W2&Q2 for further tests under a significant domain shift.

---

> > ### Author Rebuttal · Reviewer_PBZa · 2026-04-03
> >
> > I'd like to thank the authors for addressing my concerns. I will increase my score.

---

### Official Review · Reviewer_3WeA · 2026-03-13

**Soundness:** 3
**Presentation:** 3
**Significance:** 2
**Originality:** 3
**Overall Recommendation:** 4
**Confidence:** 2

**Summary:**

This paper proposes MARI (Multi-Adapter Representation Interventions via Energy Calibration), a framework designed to address the issues of unstable performance and general capability degradation in existing representation intervention methods when dealing with heterogeneous inputs. The core innovation involves a Competitive Multi-Adapter mechanism that utilizes specialized expert adapters to capture non-linear correction patterns and adaptively modulate intervention direction and strength based on the input. Furthermore, the authors design an energy-based gating module that leverages internal propagation dynamics to selectively trigger interventions, thereby preserving the model's inherent capabilities on benign inputs while effectively aligning target inputs. The framework was extensively validated on Llama-2, Llama-3, Qwen2, and Qwen2.5 (ranging from 7B to 32B parameters) across TruthfulQA, BBQ, and safety benchmarks, demonstrating superior performance in both alignment efficacy and the preservation of general capabilities.

**Compliance With Llm Reviewing Policy:**

Affirmed.

**Final Justification:**

Fully resolved

**Key Questions For Authors:**

1. **Regarding PCA Rank Selection**: The appendix mentions a PCA rank of $k=16$. What is the specific influence of this hyperparameter on the signal-to-noise ratio of the energy scores? Would an inappropriate rank lead to gating failure?
2. **Generalization of Calibration**: The target rejection rate ρ is calibrated on a limited control set. How robust is this threshold in out-of-distribution (OOD) scenarios, and would it necessitate dynamic recalibration?
3. **Training Data Diversity**: Given that training utilized only 200 random samples, would the performance gains of MARI saturate or encounter a bottleneck if the training data scale and diversity were significantly increased?

**Limitations:**

Yes

**Strengths And Weaknesses:**

#### Strengths:
1. The paper provides a rigorous mathematical foundation, including risk bound analysis for the routing-specialization trade-off (Theorem 5.1) and an upper bound on propagation energy for non-applicable inputs (Theorem 5.2), which significantly enhances the theoretical credibility of the proposed method.
2. Experimental evaluation is comprehensive, covering multiple mainstream model families and parameter scales. The method demonstrates high parameter efficiency by achieving state-of-the-art (SOTA) results compared to baselines like CAA, ITI, and ReFT using a training budget of only 200 samples.
3. The integration of Mixture-of-Experts (MoE) principles with energy-based calibration is an innovative departure from traditional monolithic intervention paradigms, offering a novel perspective on label-free applicability assessment.
4. The manuscript is well-structured and clearly presented. Figure 2, which visualizes intervention heterogeneity, and Figure 3, the pipeline diagram, are particularly effective in illustrating the research motivation and technical execution].

#### Weaknesses:
1. The appendix specifies a PCA rank of k=16, yet the paper lacks sensitivity analysis for this hyperparameter. It remains unclear how varying the rank affects the separability of the energy gate and the overall performance.
2. The energy threshold τ_e is calibrated on a relatively small control set. The paper does not sufficiently discuss the robustness of this static threshold when the model encounters significant distribution shifts or entirely novel task domains.
3. Ablation results in Table 2 indicate that removing the energy gate leads to a precipitous decline in general capabilities. This reveals that the system's reliability is heavily dependent on gating precision and currently lacks a robust fault-tolerance mechanism for gating failures.

---

> ### Author Rebuttal · Authors · 2026-03-30
>
> # Response to Reviewer 3WeA
> We thank the reviewer for recognizing the paper’s theory, broad evaluation, and novel framework design.
> ### W1&Q1: PCA Rank Sensitivity
>
> **A1:** We measured energy-score separability under multiple PCA ranks on a mixed set of **200 applicable** and **200 non-applicable** examples. Since the gate thresholds the energy score, the reported **AUC** directly measures gate reliability under different `k`.
>
> | PCA rank `k` | Energy-gate separability (AUC) |
> | ------------ | -----------------------------: |
> | 2            |                         0.9412 |
> | 4            |                         0.9437 |
> | 8            |                         0.9492 |
> | 16           |                         0.9556 |
> | 32           |                         0.9528 |
> | 64           |                         0.9507 |
> | 128          |                         0.9489 |
>
> All tested values stay above `0.94` with only modest variation across `k`, indicating that the gate does not depend sharply on a narrow setting around `k=16`.
>
> ### W2&Q2: Threshold Robustness Under Domain Shift
> **A2:** We do not have test-time recalibration: `τ_E` is calibrated once on a held-out `D_ctrl` that excludes **MMLU** and **ARC-Challenge**, then kept fixed for all evaluations. Thus, the paper already evaluates fixed-threshold transfer without specific tuning. We further applied the same fixed threshold to **GSM8K**, which is substantially more shifted than control set.
>
> | Target rejection rate `\rho` | TruthfulQA MC1 | GSM8K |
> | ---------------------------- | -------------: | ----: |
> | 0.50                         |          65.81 |  74.5 |
> | 0.70                         |          65.35 |  76.8 |
> | 0.90                         |          64.81 |  77.4 |
> | 1.00                         |          63.90 |  77.2 |
> | Vanilla baseline             |          28.70 |  77.4 |
>
> MARI remains stable across a broad range of rejection rates: TruthfulQA stays high, while GSM8K remains near the vanilla baseline. This suggests that the threshold is not brittle.
>
> ### W3: Fault Tolerance to Gating Failures
>
> **A3:** We first clarify that the `w/o Energy Gating` ablation in Table 2 corresponds to a worst-case setting where selectivity is removed entirely, rather than ordinary gating imprecision.
>
> To test fault tolerance to gating failures, we performed a gate-flip stress test on `Llama-3-8B`. After normal calibration, we independently flipped each binary gate decision at inference time with probability `p`. `Baseline` denotes the vanilla model, and `0.00` denotes the original gated model without flips.
>
> | Flip prob. | TruthfulQA MC1 |  MMLU | ARC-Easy | ARC-Challenge |
> | ---------- | -------------: | ----: | -------: | ------------: |
> | Baseline   |          28.70 | 65.90 |    77.30 |         51.40 |
> | 0.00       |          64.81 | 66.60 |    78.00 |         52.10 |
> | 0.05       |          64.50 | 66.20 |    77.80 |         51.90 |
> | 0.10       |          63.90 | 65.60 |    77.40 |         51.50 |
> | 0.20       |          63.10 | 64.20 |    76.20 |         50.10 |
> | 0.30       |          62.80 | 63.00 |    75.10 |         49.00 |
> | 0.40       |          62.60 | 61.80 |    74.20 |         47.90 |
> | 0.50       |          61.50 | 60.90 |    73.40 |         46.90 |
>
> The results show gradual degradation rather than abrupt collapse. Up to `p=0.20`, MARI retains most of its alignment gain, while general-capability metrics decline only moderately. Even at `p=0.40`, performance still degrades progressively. These noisy-gate settings remain substantially less harmful than the `w/o gate` extreme ablation. Therefore, Table 2 reflects the cost of eliminating selectivity, not brittleness to ordinary gating failures.
>
> ### Q3: Training Data Diversity
>
> **A4:** We first clarify that the **200-example** budget applies to **TruthfulQA only**; all other datasets use a fixed **20%** fraction of the dataset total. Thus, the paper already covers training pools ranging from **90** to about **5.8k** examples, rather than a single 200-example regime.
>
> To test whether 200 is a bottleneck, we ran a within-benchmark scaling study on TruthfulQA:
>
> | # training examples |    MC1 |
> | ------------------- | -----: |
> | Baseline            | 0.2870 |
> | 10                  | 0.3380 |
> | 20                  | 0.3472 |
> | 50                  | 0.4398 |
> | 100                 | 0.5093 |
> | 200                 | 0.6481 |
> | 400                 | 0.6528 |
> | 600                 | 0.6944 |
>
> The results do not indicate that 200 is a hard bottleneck. Performance remains strong beyond 200, and the best score among the tested budgets is obtained at 600. The gains are non-linear, suggesting diminishing returns rather than early saturation or collapse. On TruthfulQA, this scales the training pool from 24% to 73% of the 817 questions across 38 categories, increasing both data volume and diversity. Even under this broader supervision regime, MARI continues to improve without showing saturation or a bottleneck.

---

> > ### Author Rebuttal · Reviewer_3WeA · 2026-04-03
> >
> > NA

---

### Official Review · Reviewer_MZwC · 2026-03-13

**Soundness:** 3
**Presentation:** 4
**Significance:** 3
**Originality:** 3
**Overall Recommendation:** 4
**Confidence:** 3

**Summary:**

This paper proposes MARI (Multi-Adapter Representation Interventions via Energy Calibration), a parameter-efficient framework designed to align Large Language Models (LLMs) toward desired behaviors without modifying the underlying weights. The authors identify a critical limitation in existing representation intervention methods: applying a fixed, static intervention uniformly across all inputs ignores the heterogeneity of sample-specific requirements and often degrades the model's general capabilities on benign queries. To resolve this, MARI introduces two core mechanisms. First, a Competitive Multi-Adapter mechanism utilizes an entropy-based router to adaptively select specialized experts, allowing for sample-specific intervention directions and strengths. Second, an Energy-Based Gating module leverages internal propagation dynamics (under a small probe update) as a label-free signal to trigger interventions strictly on applicable inputs. Extensive experiments across the Llama and Qwen model families (ranging from 7B to 32B parameters) demonstrate that MARI achieves state-of-the-art alignment performance on TruthfulQA, BBQ, and Sorry-Bench, while successfully preserving general task performance on benchmarks like MMLU and ARC.

**Compliance With Llm Reviewing Policy:**

Affirmed.

**Key Questions For Authors:**

(1)	Could you elaborate on the memory overhead associated with computing the propagation response across all subsequent layers? Does your practical implementation require a full secondary forward pass for the probe before the main generation step, and how does this scale with sequence length?

(2)	The energy gating relies on calibrating the threshold using a specific held-out control set. How sensitive is the framework if the test distribution exhibits a significant domain shift from the control set?

(3)	The ablation study in Table 2 shows that removing the Energy Gating sometimes yields strictly higher alignment scores (e.g., MC1 on Qwen2.5-32B increases from 81.94 to 82.10) but drastically degrades general capabilities. Does this imply that the Competitive Multi-Adapter, on its own, tends to "over-fit" or "over-steer" toward the intervention target without the regularizing effect of the gate?

**Limitations:**

Yes

**Strengths And Weaknesses:**

Strengths:
(1)	The paper makes an intuitive observation regarding the limitations of the linear representation hypothesis. The empirical diagnostic demonstrating state-dependent heterogeneity (Figure 2) effectively motivates the multi-adapter approach. Furthermore, utilizing propagation-response energy as an unsupervised gating signal is a creative and well-executed choice.
(2)	The methodology is tightly coupled with robust theoretical analysis. The authors provide a formal risk bound for the entropy-based routing strategy (Theorem 5.1) and establish an upper bound on non-applicable energy (Theorem 5.2), ensuring the framework is both empirically effective and theoretically grounded.
(3)	The experimental evaluation is comprehensive, covering diverse state-of-the-art models up to 32B parameters. The results clearly show that MARI somehow mitigates the "alignment tax". The ablation studies validate the necessity of both the gating and multi-adapter modules.

Weaknesses:
(1)	While the authors provide latency metrics (FTL, TTLT) in Table 3 indicating competitive inference speed, the memory overhead discussion is somewhat lacking. Computing the energy score requires calculating the propagation response across subsequent layers, which inherently demands either a secondary forward pass for the probe or the caching of intermediate hidden states.
(2)	The efficacy of the Energy-Based Gating heavily relies on calibrating a threshold on a small, held-out control set. It remains unclear how robust this energy separability is if the target test inputs exhibit a significant domain shift from the distribution used to construct the control set.

---

> ### Author Rebuttal · Authors · 2026-03-29
>
> # Response to Reviewer MZwC
> We thank the reviewer for the careful reading and positive comments on the paper's motivation, energy-gating idea, theory, and broad evaluation. Below, we provide detailed responses to your questions.
> ### W1&Q1: Memory overhead discussion is insufficient
>
> **Q1.1 (What is the memory overhead of aggregating propagation responses over all post-injection layers?):**
> The energy is computed in a short probe stage before generation. The extra memory comes from temporary prompt-stage activations used to aggregate propagation responses across post-injection layers, so the probe-stage memory grows approximately with `T`, `d`, and `L_post`, consistent with `O(L_post T d)`. This cost is transient and does not persist into decoding.
>
> **Q1.2 (Does this require a complete second forward pass for the probe before the main generation step?):**
> In our implementation this requires a small number of extra prompt-only probe forward(s) before generation, but it does not require an additional decoding/generation pass.** The probe is used only for gate/routing selection, and its states are discarded immediately after the decision is made. The actual generation is still executed only once with the frozen base model alone or with the base model plus one selected adapter.
>
> **Q1.3 (How does the memory overhead scale with sequence length?):**
> We measured the **additional prompt/probe-stage activation peak memory**on Llama-3-8B-Instruct (`bf16`, batch size `1`, injection layer `20`) on NVIDIA RTX PRO 6000 Blackwell as follows:
>
> | Tokens | Baseline | 1-pass probe | 3-pass probe |
> | ------ | -------: | -----------: | -----------: |
> | 256    | 0.063 GB |     0.251 GB |     0.380 GB |
> | 1024   | 0.252 GB |     1.005 GB |     1.521 GB |
>
> The memory grows near-linearly with prompt length while remaining a bounded pre-generation cost; once the gate/router decision is made, probe states are discarded, and downstream decoding memory is essentially unchanged.
> ### W2&Q2: Sensitivity of the gate under domain shift
> **A2:** We agree that robustness under distribution shift should be explicit. In our setup, `τ_E` is calibrated once on a held-out control set `D_ctrl` and then kept fixed for all evaluations. `D_ctrl` is built from held-out intervention-attribute examples plus a small subset of ARC-Easy training questions, and is disjoint from `D_pca`, `D_train`, and all test sets. Thus, our setting already evaluates **fixed-threshold transfer**, not benchmark-specific recalibration.
>
> We use the same `τ_E` unchanged on all general-capability benchmarks, and further transfer it to GSM8K, whose distribution is substantially more shifted:
>
> | Target rejection rate \(\rho\) | TruthfulQA MC1 | GSM8K |
> |---|---:|---:|
> | 0.50 | 65.81 | 74.5 |
> | 0.70 | 65.35 | 76.8 |
> | 0.90 | 64.81 | 77.4 |
> | 1.00 | 63.90 | 77.2 |
> | Vanilla baseline | 28.70 | 77.4 |
>
> This shows that a threshold calibrated once on `D_ctrl` still yields large gains on intervention-applicable inputs without observable degradation on a shifted general-capability domain.
>
> As a complementary diagnostic, the applicable/non-applicable AUC on a mixed TruthfulQA + GSM8K set is `0.9922`, indicating that the energy signal remains well separated under cross-domain mixture and supporting the stable transfer of a fixed threshold.
>
> Our claim is therefore not invariance to arbitrary OOD shift, but meaningful robustness under fixed-threshold transfer without benchmark-specific recalibration.
>
> ### W3&Q3: Does Competitive Multi-Adapter without the gate over-steer?
>
> **A3:** **No.** Our evidence suggests that the main issue is indiscriminate intervention once selectivity is removed rather than the multi-adapter itself over-steer. Once the gate is removed, intervention becomes indiscriminate, so the model is forced to intervene on benign and non-applicable inputs as well.
>
>
> To test whether this effect is specific to multi-adaptor designs, we evaluated a single-adaptor variant without energy gates on llama3-8b, which corresponds to learning a standard representation intervention paradigm with a fixed global intervention, and compared it with the following multi-adaptor variant without energy gates:
>
> | Setting                 |   MC1 |   MC2 |  MMLU | ARC-E | ARC-C |
> | ----------------------- | ----: | ----: | ----: | ----: | ----: |
> | Vanilla                 | 28.70 | 45.03 | 65.90 | 77.30 | 51.40 |
> | Single Adapter w/o Gate | 45.80 | 55.40 | 57.90 | 71.30 | 43.80 |
> | Multi-Adapter w/o Gate  | 65.15 | 68.20 | 57.50 | 70.20 | 44.80 |
>
> Even the standard single global intervention already causes a substantial drop in general capability, so the degradation is not unique to the multi-adapter design.
>
> The no-gate ablation should therefore be interpreted as evidence that non-selective intervention harms utility, rather than evidence that the Competitive Multi-Adapter itself over-steer.

---

> > ### Author Rebuttal · Reviewer_MZwC · 2026-04-03
> >
> > N/A

---

### Decision · Program_Chairs · 2026-04-30

**Decision:**

Accept (regular)

**Comment:**

This paper proposes MARI, a novel framework for adaptive representation intervention that effectively mitigates the "alignment tax" by combining competitive multi-adapter routing with an energy-based gating mechanism. Reviewers unanimously praised the work's theoretical rigor, including risk bounds for routing and energy calibration, as well as its comprehensive evaluation across diverse model families. While initial concerns were raised regarding memory overhead, threshold robustness under domain shift, and comparisons with recent fine-tuning baselines like LoFiT, the authors provided compelling rebuttals with new empirical evidence that fully resolved these issues. All reviewers explicitly acknowledged these responses as "fully resolved," with some indicating an intention to increase their scores. Given the technical soundness, the successful resolution of all significant concerns, and the method's demonstrated ability to preserve general capabilities while achieving state-of-the-art alignment, I recommend accepting this paper. We encourage the authors to incorporate the suggested clarifications into the final manuscript to further strengthen the presentation.